# Smooth muscle FGF/TGFβ cross talk regulates atherosclerosis progression

Pei-Yu Chen[1,*,†], Lingfeng Qin[2,†], Guangxin Li[2,3,†], George Tellides[2] & Michael Simons[1,4,**]

## Abstract

The conversion of vascular smooth muscle cells (SMCs) from contractile to proliferative phenotype is thought to play an important role in atherosclerosis. However, the contribution of this process to plaque growth has never been fully defined. In this study, we show that activation of SMC TGFβ signaling, achieved by suppression of SMC fibroblast growth factor (FGF) signaling input, induces their conversion to a contractile phenotype and dramatically reduces atherosclerotic plaque size. The FGF/TGFβ signaling cross talk was observed *in vitro* and *in vivo*. *In vitro*, inhibition of FGF signaling increased TGFβ activity, thereby promoting smooth muscle differentiation and decreasing proliferation. *In vivo*, smooth muscle-specific knockout of an FGF receptor adaptor *Frs2α* led to a profound inhibition of atherosclerotic plaque growth when these animals were crossed on *Apoe*$^{−/−}$ background and subjected to a high-fat diet. In particular, there was a significant reduction in plaque cellularity, increase in fibrous cap area, and decrease in necrotic core size. In agreement with these findings, examination of human coronary arteries with various degrees of atherosclerosis revealed a strong correlation between the activation of FGF signaling, loss of TGFβ activity, and increased disease severity. These results identify SMC FGF/TGFβ signaling cross talk as an important regulator of SMC phenotype switch and document a major contribution of medial SMC proliferation to atherosclerotic plaque growth.

**Keywords** atherosclerosis; FGF/TGFβ; smooth muscle cells
**Subject Categories** Cardiovascular System; Vascular Biology & Angiogenesis

## Introduction

In healthy mature blood vessels, vascular smooth muscle cells (SMCs) are quiescent, fully differentiated cells that exhibit a very low rate of proliferation. They express a number of contractile proteins necessary for maintaining vessel tone, blood pressure, and blood flow, including smooth muscle α-actin (SM α-actin), smooth muscle 22 alpha (SM22α), SM-calponin, and smooth muscle myosin heavy chain (SM-MHC) (Owens *et al*, 2004; Shi & Chen, 2014; Liu *et al*, 2015). Following vascular injury or in association with a variety of diseases, SMCs exhibit a decrease in expression of differentiation markers and acquire a proliferative phenotype characterized by enhanced cell proliferation and migration (Owens *et al*, 2004; Kawai-Kowase & Owens, 2007). This form of SMC phenotypic modulation is especially robust in atherosclerosis and vascular stenosis following angioplasty where it is thought to contribute to the growth of atherosclerotic plaques and neointima (Marx *et al*, 2011; Gomez & Owens, 2012; Tabas *et al*, 2015). Therefore, elucidation of mechanisms that control normal SMC phenotypic switch in disease states is likely to provide key insights toward understanding the biology of atherosclerosis and development of new therapeutic targets.

Smooth muscle differentiation is promoted by a number of signaling pathways including transforming growth factor β (TGFβ), Notch3 as well as integrin- and extracellular matrix-derived differentiation signals. TGFβ signaling is particularly critical for the maintenance of normal adult vasculature (Li *et al*, 2014) and the growth factor plays a critical role in mediating balance between inflammation and fibrous plaque growth in atherosclerosis (Lutgens *et al*, 2002). TGFβ exerts its effects via a complex of two serine/threonine kinase type II receptors (TGFβRII) and the type I receptor Alk5 (TGFβRI) (Carvalho *et al*, 2007; Mack, 2011). TGFβRI phosphorylation by TGFβRII results in recruitment and phosphorylation of Smad2 and Smad3 that then complex with Smad4 and translocate to the nucleus. Subsequent activation of contractile SMC-specific gene expression involves both direct binding of Smads to certain DNA binding sites and interactions with other SMC transcription factors such as SRF and myocardin. TGFβ also activates non-Smad-dependent signaling pathways that also play a role in the induction of SMC differentiation (Li *et al*, 2014). In agreement with these results, genetic deletions of either TGFβ1, TGFβ2, their receptors (TGFβR1, TGFβR2), or signaling molecules (Smad2, Smad3) are all associated with various vascular wall pathologies including aneurysm formation (Carvalho *et al*, 2007; Tang *et al*, 2010; Doyle *et al*, 2012; Lindsay *et al*, 2012; Li *et al*, 2014; Crosas-Molist *et al*, 2015).

1   Department of Internal Medicine, Yale Cardiovascular Research Center, Yale University School of Medicine, New Haven, CT, USA
2   Department of Surgery, Yale University School of Medicine, New Haven, CT, USA
3   Department of Vascular Surgery, The First Hospital of China Medical University, Shenyang, China
4   Department of Cell Biology, Yale University School of Medicine, New Haven, CT, USA
    *Corresponding author. Tel: +1 203 737 4643; Fax: +1 203 737 5528; E-mail: pei-yu.chen@yale.edu
    **Corresponding author. Tel: +1 203 737 4643; Fax: +1 203 737 5528; E-mail: michael.simons@yale.edu
    †These authors contributed equally to this work

While the central role played by TGFβ in regulation of SMC differentiation has been previously demonstrated (Lindner & Reidy, 1991; Hirschi *et al*, 1998; Kawai-Kowase *et al*, 2004), little is known about what regulates this pathway and what contribution SMC proliferation makes to progression of lesions seen in atherosclerosis (Tabas *et al*, 2015). Recent studies in endothelial cells demonstrated fibroblast growth factor (FGF)-dependent regulation of TGFβ. The loss of endothelial cell FGF signaling input *in vitro* or *in vivo* leads to a profound decrease in *let-7* miRNA levels that results in marked prolongation of TGFβR1 mRNA half-life and increased TGFβR1 protein expression. Together with a large increase in TGFβ2 levels, this leads to activation of TGFβ signaling including phosphorylation of Smad2 and Smad3 and induction of expression of various smooth muscle and mesenchymal markers, thereby inducing endothelial-to-mesenchymal transition (EndMT) (Chen *et al*, 2012, 2014). Importantly, EndMT, in turn, leads to acceleration of atherosclerosis progression (Chen *et al*, 2015).

Prior studies also reported FGF antagonism of TGFβ activity in SMCs and pericytes *in vitro*, but the mechanism of this effect and its functional consequences have not been fully established (Papetti *et al*, 2003; Kawai-Kowase *et al*, 2004). We hypothesized suppression of FGF signaling in SMC would induce a contractile phenotype and that this enforced maintenance of contractile SMC phenotype would diminish any contributions of medial smooth muscle cell proliferation to atherosclerotic plaque growth.

To investigate this hypothesis, we generated a mouse line with an SMC-specific deletion of a key FGF signaling regulator fibroblast growth factor receptor substrate 2 alpha (*Frs2α*). The shutdown of FGF-induced MAPK signaling in SMCs induced by *Frs2α* knockout resulted in increased expression of TGFβ ligands and receptors and activation of TGFβ signaling. *In vitro*, this led to a growth arrest of proliferating SMCs and induction of their differentiation, while

*in vivo*, there was a profound reduction in the size of atherosclerotic lesions. Analysis of clinical specimens confirmed the inverse relationship between the extent of medial FGF and TGFβ signaling and the severity of atherosclerosis.

Overall, these results demonstrate that FGF regulates SMC phenotypic modulation by controlling SMC TGFβ signaling and directly elucidate the contribution of SMC proliferation to the growth of atherosclerotic plaque.

## Results

### FRS2α regulates TGFβ activity and SMC differentiation

We first examined whether inhibition of FGF signaling in SMCs using FRS2α knockdown affects the expression of TGFβ pathway signaling molecules. In cultured human aortic smooth muscle cells (HASMCs), knockdown of FRS2α led to a significant increase in expression of TGFβ2, TGFβ3, TGFβR1, and TGFβR2 (Fig 1A). TGFβ1 was unchanged. In addition, there was an increase in the expression of a number of TGFβ-dependent genes including connective tissue growth factor (CTGF), elastin, plasminogen activator inhibitor-1 (PAI-1), p21, p27, and collagen (Fig 1B) suggesting activation of TGFβ signaling. This was confirmed by Western blotting that demonstrated increased phosphorylation of Smad2 and Smad3 following FRS2α knockdown (Fig 1C).

Cultured SMCs in serum-supplemented medium lose differentiation marker expression and acquire a synthetic (proliferative) phenotype. Since activation of TGFβ signaling has been linked with the induction of SMC differentiation, we next examined differentiation marker expression in cultured HASMC following FRS2α knockdown. There was a pronounced increase in expression of SM

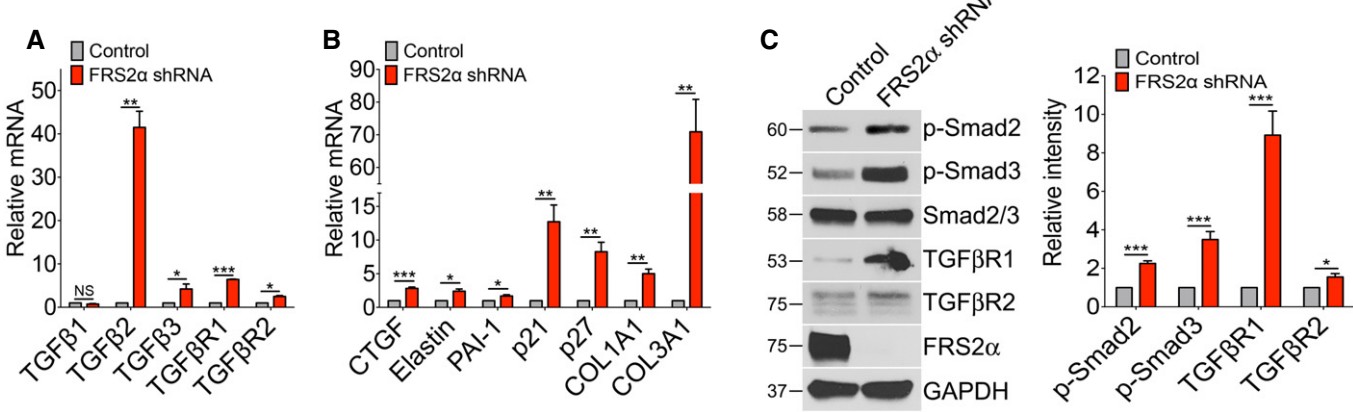

**Figure 1. FRS2α knockdown activates TGFβ signaling in primary human aortic smooth muscle cells (HASMCs).**

A, B  qRT–PCR analysis of TGFβ ligands, TGFβ receptors, and TGFβ target expression in control and FRS2α-knockdown HASMCs. β-actin was used for sample loading normalization. Histogram of qRT–PCR results is representative of three independent experiments.

C  Left: Immunoblot analysis of TGFβRs, phosphorylated Smad2 (p-Smad2), and phosphorylated Smad3 (p-Smad3) in control and FRS2α-knockdown HASMCs. Blots are representative of four independent experiments. Right: Band intensities of p-Smad2, p-Smad3, TGFβR1, and TGFβR2 were normalized to Smad2/3 or GAPDH and expressed as a fraction of a control value.

Data information: Results are expressed as means ± SD (NS: not significant compared to control; *P < 0.05, **P < 0.01, ***P < 0.001 compared to control; unpaired two-tailed Student's *t*-test).

Source data are available online for this figure.

α-actin, SM22α, and SM-calponin (Fig 2A) as well as various transcription factors (GATA6, MyoCD, SRF) and transcription co-activators (MKL1, MKL2) responsible for the induction of contractile phenotype (Fig 2B). The contractile machinery was functional as observed by increased contraction of collagen gels following FRS2α knockdown (Fig 2C).

To assess the effect of FGF signaling shutdown on SMC proliferation, real-time cell analysis was used to track HASMC growth in the presence and absence of FRS2α knockdown. The absence of FRS2α expression resulted in nearly complete inhibition of serum-induced HASMC proliferation (Appendix Fig S1A). Western blot analysis demonstrated a decrease in the proliferative marker cyclin D1, whereas expression of cell cycle inhibitor proteins p21 and p27 was upregulated (Appendix Fig S1B; source data for full unedited gels are available

online). In agreement with these findings, FACS analysis showed a G1/S arrest following FRS2α knockdown (Appendix Fig S1C).

To test whether TGFβ activity is required for FRS2α-knockdown-induced SMC differentiation, HASMCs were exposed to FRS2α or control shRNA lentiviruses in the presence or absence of the TGFβR1 kinase inhibitor, SB431542. The inhibitor treatment effectively attenuated FRS2α-knockdown-induced increase in SM-calponin and p-Smad2 levels (Fig 2D) demonstrating that TGFβ activity is essential for FRS2α-knockdown-induced contractile smooth muscle gene expression. This was further confirmed by shRNA-mediated knockdown of TGFβR2 or Smad2 with both knockdowns preventing increase in SM-calponin expression (Fig 2E and F). Importantly, inhibition of TGFβ signaling using a variety of means (TGFβR1 inhibitor SB431542, TGFβR2 shRNA, and Smad2

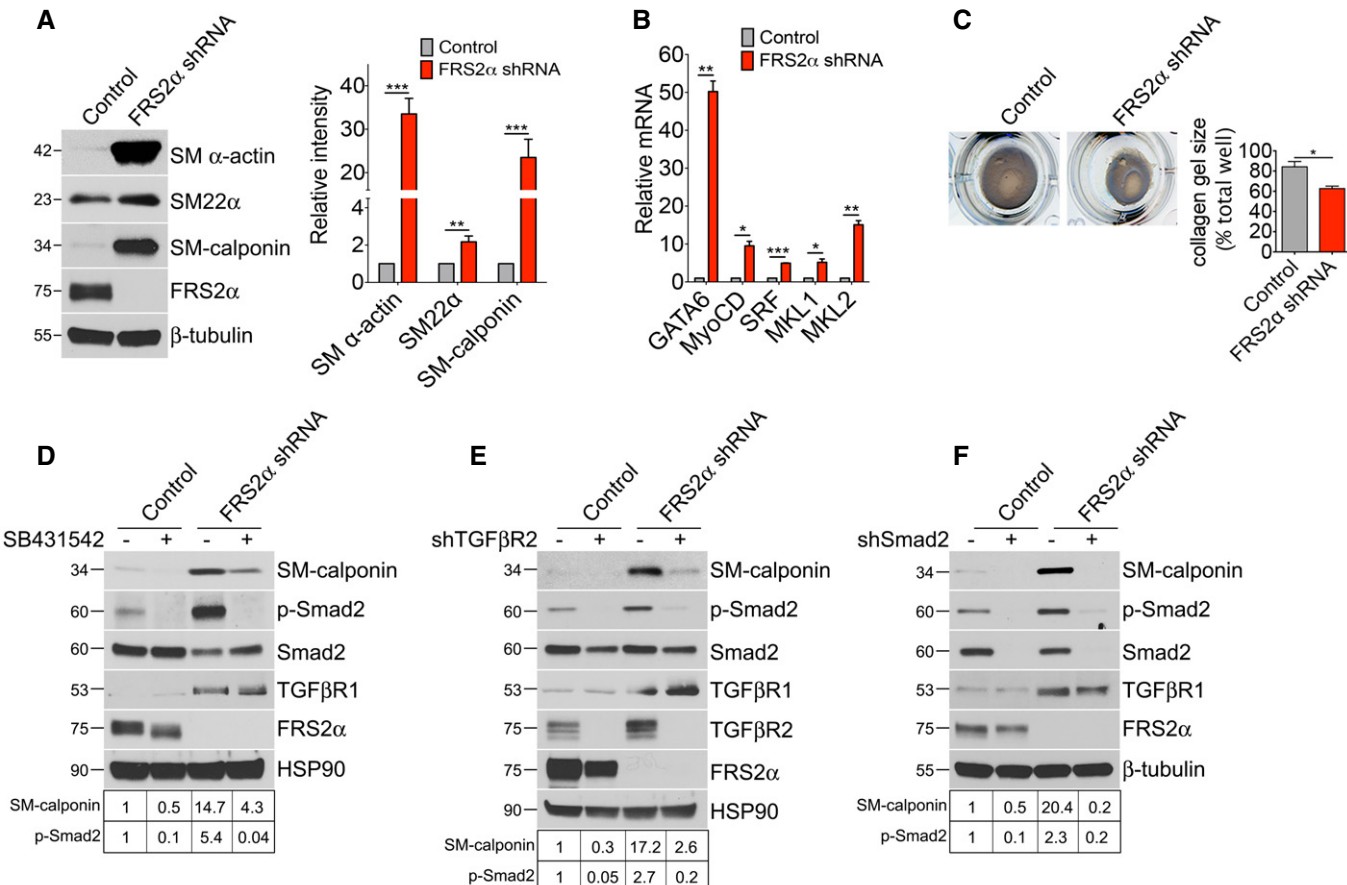

**Figure 2. FRS2α knockdown increases smooth muscle marker gene expression via the TGFβ pathway in primary human aortic smooth muscle cells (HASMCs).**

A    Left: Immunoblot analysis of smooth muscle marker gene expression in control and FRS2α-knockdown HASMCs. Blots are representative of four independent experiments. Right: Band intensities of SM α-actin, SM22α, and SM-calponin were normalized to β-tubulin and expressed as a fraction of a control value.

B    qRT–PCR analysis of SMC transcription factor gene expression in control and FRS2α-knockdown HASMCs. β-actin was used for sample loading normalization. Histogram of qRT-PCR results is representative of three independent experiments.

C    Collagen gel contraction assays were used to determine the contractile ability of control or FRS2α-knockdown HASMCs. Histogram of collagen gel contraction assays is representative of three independent experiments.

D–F    Upper panels: Immunoblots of smooth muscle markers, phosphorylated Smad2 (p-Smad2), and TGFβR1 expression in control and FRS2α-knockdown HASMCs treated with SB431542 (10 μM), TGFβR2, or Smad2 shRNA lentiviruses. Blots are representative of three independent experiments. Bottom panels: Band intensities of SM-calponin and p-Smad2 were normalized to β-tubulin, HSP90, or Smad2 and expressed as a fraction of a control value.

Data information: Results are expressed as means ± SD (*P < 0.05, **P < 0.01, ***P < 0.001 compared to control; unpaired two-tailed Student's *t*-test).
Source data are available online for this figure.

shRNA) resulted in partial reversal of FRS2α-knockdown-induced growth arrest (Fig EV1).

### FGFR1 and *let-7* mediate FGF-driven suppression of TGFβ signaling in SMCs

We previously showed that suppression of FGF signaling in endothelial cells decreases expression of *let-7* miRNA family members (Chen *et al*, 2012, 2014). To assess whether the same mechanism is operational in SMCs, *let-7* levels were examined after shRNA-mediated FRS2α knockdown in HASMCs. As in endothelial

cells, this led to a substantial decrease in *let-7* miRNA expression in FRS2α-knockdown HASMCs (Fig 3A). Transduction of *let-7*b lentiviruses into HASMCs following FRS2α knockdown prevented activation of TGFβ signaling as demonstrated by decreased TGFβR1, p-Smad2, and SM-calponin levels (Fig 3B).

Growth arrest of cultured SMCs has been shown to induce their conversion from proliferative-to-contractile phenotype (Clowes *et al*, 1988). Indeed, shifting HASMC cultured in 4.9% FBS to 1% FBS medium led to a gradual increase in expression of contractile SMC proteins (Fig 3C). Analysis of *let-7* family members' expression during HASMC differentiation demonstrated a profound decrease

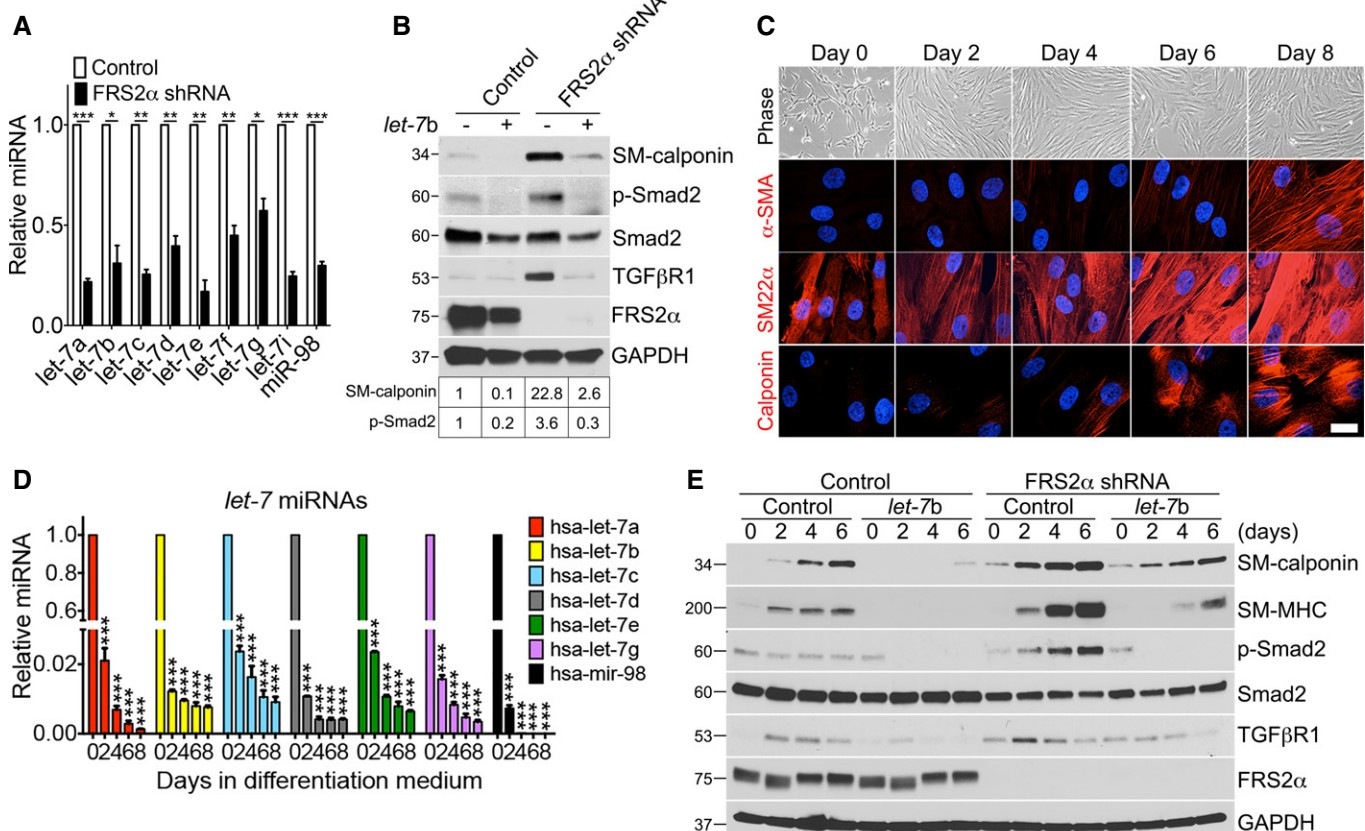

**Figure 3. FRS2α knockdown increases smooth muscle marker gene expression via the *let-7*-TGFβ pathway in primary human aortic smooth muscle cells (HASMCs).**

A   Quantitative real-time PCR analysis of mature *let-7* family in control and FRS2α-knockdown HASMCs. SNORD47 was used to normalize the variability in template loading. Histogram of qRT–PCR results is representative of three independent experiments.

B   Upper panel: Immunoblots of SM-calponin, phosphorylated Smad2 (p-Smad2), and TGFβR1 expression in control and FRS2α-knockdown HASMCs transduced with or without *let-7*b lentiviruses. Blots are representative of three independent experiments. Bottom panels: Band intensities of SM-calponin and p-Smad2 were normalized to GAPDH or Smad2 and expressed as a fraction of a control value.

C   Phase-contrast and immunofluorescence staining of smooth muscle markers (red) in HASMCs. Nuclei were counterstained with DAPI (blue). Scale bar: 12 μm. Images are representative of three independent experiments.

D   HASMCs were cultured in the growth medium (M231+ SMGS) at day 0 and then switched from growth conditions to differentiation medium (M231+ SMDS) for 8 days. Quantitative real-time PCR analysis of mature *let-7* family in HASMCs. SNORD47 was used to normalize the variability in template loading. Histogram of qRT–PCR results is representative of three independent experiments.

E   Control and FRS2α-knockdown HASMCs were cultured in the growth medium (M231+ SMGS) at day 0 and then switched from growth conditions to differentiation medium (M231+ SMDS) for 6 days with or without *let-7*b lentiviruses. Immunoblots of smooth muscle markers, phosphorylated Smad2 (p-Smad2), and TGFβR1 expression in control and FRS2α-knockdown HASMCs with or without *let-7*b lentiviruses. Blots are representative of three independent experiments.

Data information: Results are expressed as means ± SD; **$P < 0.01$, ***$P < 0.001$ compared to control; unpaired two-tailed Student's *t*-test (A) or one-way ANOVA with Newman–Keuls *post hoc* test for multiple comparison correction (D).
Source data are available online for this figure.

that preceded changes in contractile proteins expression suggesting *let-7*-dependent control of this process (Fig 3D).

To test this further, HASMCs shifted to the growth arrest medium were exposed to FRS2α or control shRNA lentiviruses in the presence or absence of the *let-7*b lentivirus. In agreement with the data presented above, HASMC FRS2α knockdown accelerated reversion to the contractile phenotype (Fig 3E). The phenotype conversion, however, was effectively blocked by *let-7* overexpression as demonstrated by decreased TGFβR1, SM-calponin, and SM-MHC expression and reduced Smad2 phosphorylation (Fig 3E).

Since FRS2α is involved in signaling of all four FGF receptors, we next set out to determine the principle FGFR responsible for suppression of TGFβ signaling in SMC. qPCR analysis demonstrated that FGFR1 was the main FGFR expressed in cultured HASMCs (Appendix Fig S2A). In agreement with that finding, shRNA-mediated FGFR1 knockdown markedly increased TGFβ2, TGFβ3, TGFβR1, and TGFβR2 expression (Appendix Fig S2B) in a manner similar to that of the FRS2α knockdown. This also led to activation of TGFβ signaling as demonstrated by increased expression of a number of TGFβ-dependent genes and transcription factors (Appendix Fig S2C and D). Western blotting confirmed activation of TGFβ signaling as demonstrated by increased Smad2 and Smad3 phosphorylation and increased contractile SMC gene expression (Appendix Fig S2E; source data for full unedited gels are available online). Finally, inhibition of TGFβ signaling (SB431542, TGFβR2 shRNA, and Smad2 shRNA) in growth condition (Fig EV2A–C) or overexpression of *let-7*b lentiviruses in differentiation condition (Fig EV2D and E) was able to reverse FGFR1-knockdown-induced SMC contractile phenotype.

### Activation of FGF and loss of TGFβ signaling in human and mouse atherosclerotic lesions

To examine the role played by FGF regulation of TGFβ signaling activity in SMCs in disease settings, we first evaluated the correlation between medial FGF and TGFβ signaling and the severity of atherosclerosis in samples of left main coronary arteries from forty-three patients (Fig 4A and B). Table 1 summarizes clinical characteristics of this patient group. Immunostaining of serial left main coronary artery sections for SM α-actin and SM-MHC revealed decreased expression of these contractile SMC markers in the media of arteries from patients with moderate and severe coronary

atherosclerosis compared to patients with no/mild disease (Fig 4C and D), consistent with previous findings (Glukhova *et al*, 1988; Aikawa *et al*, 1993, 1995). At the same time, there was an increase in immunoreactivity for the phosphorylated form of FGFR1 in patients with moderate and severe coronary artery disease (CAD), implying an increase in FGF signaling (Fig 4E and F). Yet, there was no change in the medial FGFR1 expression levels (Fig 4G and H).

This activation of FGF signaling and the loss of smooth muscle contractile markers in advanced atherosclerotic lesions was accompanied by a decrease in TGFβ immunoreactivity in the media and the loss of p-Smad2 and p-Smad3 expression (Fig 5A–F). Quantification of immunocytochemistry data from the left main coronary arteries of the entire patient cohort showed that while 84% of SMCs in patients with no/mild CAD demonstrated expression of p-Smad2 in the media of their coronary arteries, this was reduced to 21% in patients with moderate CAD and 6% in patients with severe CAD (Fig 5D). Similarly, 83% of SMCs in patients with no/mild CAD demonstrated expression of p-Smad3 in the media of their coronary arteries; this was reduced to 41% in patients with moderate CAD and 16% in patients with severe CAD (Fig 5F).

These findings were confirmed in an $Apoe^{-/-}$ mouse model of atherosclerosis. After 16 weeks of high-fat diet (HFD), medial SMCs in brachiocephalic artery atherosclerotic plaque had decreased expression of contractile SMC proteins compared to medial SMC of mice on a normal chow diet (Fig 6A and B). This correlated with increased SMC p-FGFR1 expression in the media (Fig 6C and G), while total FGFR1 levels were unchanged (Fig 6D and H) and decreased p-Smad2 and p-Smad3 activities (Fig 6E, F, I and J).

### Smooth muscle-specific *Frs2α* deletion reduces atherosclerotic lesion growth

To further study the link between the loss of SMC FGF signaling and their phenotype modulation *in vivo*, we generated mice with an SMC-specific *Frs2α* deletion ($Frs2α^{SMCKO}$) using the SM22αCre line (Holtwick *et al*, 2002). $Frs2α^{SMCKO}$ mice were viable and born at the expected Mendelian frequency. Assessment of FRS2α expression levels in vascular tissue revealed a robust deletion of FRS2α in the aorta (Fig EV3A–C). There were no differences in the gross appearance of ascending or descending aorta between control and $Frs2α^{SMCKO}$ mice (Fig EV3D) nor was there any difference in arterial wall thickness (elastic Van Gieson staining), smooth muscle

**Figure 4. FGFR1 signaling activity in smooth muscle cells in human left main coronary arteries with various degrees of atherosclerosis.**

A    Coronary arteries dissected from the human heart. Left main (LM), left anterior descending (LAD), and left circumflex (LCX) branches. Scale bar: 1 cm.
B    Elastic Van Gieson (EVG) staining of human coronary arteries demonstrating various degrees of atherosclerosis.
C, D    Representative images of immunofluorescence staining for CD31 (green) and SM α-actin (red) or SM-MHC (red) in no/mild, moderate, and severe disease human left main coronary arteries. No: no disease. Nuclei were stained with DAPI (blue). Panels b, d, f are high-magnification view from a, c, e. Images are representative of 10 no/mild, 9 moderate, and 10 severe disease human left main coronary artery samples. L: lumen. Scale bar: 16 μm.
E    Representative images of immunofluorescence staining for p-FGFR1 (red) in the same patient cohort. Nuclei were counterstained with DAPI (blue). Scale bar: 16 μm.
F    Percentage of medial p-FGFR1⁺ SMC (***$P < 0.001$ compared to no/mild disease; one-way ANOVA with Newman–Keuls *post hoc* test for multiple comparison correction).
G    Representative images of immunofluorescence staining for FGFR1 (red) in the same patient cohort. Nuclei were counterstained with DAPI (blue). Scale bar: 16 μm.
H    Percentage of medial FGFR1⁺ SMC (NS: not significant compared to no/mild disease; one-way ANOVA with Newman–Keuls *post hoc* test for multiple comparison correction).

Data information: The data shown in (F, H) are the means ± SD of the percentage of medial p-FGFR1 or FGFR1-positive SMC. Images are representative of 10 No/mild, 9 moderate, and 10 severe disease human left main coronary artery samples. A full table of *P*-values for this figure is shown in Appendix Table S1.

    

contractile marker gene expression (SM α-actin, SM22α, Notch3), phosphorylated Smad2 (p-Smad2), and vascular density in the heart and skeletal muscle (Fig EV3E–H). Thus, the deletion of FRS2α *per se* did not alter the baseline structure of the normal vasculature.

To study the role of FGF signaling in the modulation of SMC phenotype during atherogenesis, we crossed *Frs2α^SMCKO* mice

onto the atherosclerosis-prone *Apoe^−/−* background (*Frs2α^SMCKO*/ *Apoe^−/−*). Male *Frs2α^SMCKO*/*Apoe^−/−* and *Apoe^−/−* littermates were placed on cholesterol-rich Western diet for 8 or 16 weeks at which point whole-mount Oil Red O staining was used to visualize the extent of aortic atherosclerotic plaques. There were no differences between these two groups with regard to body weight, total

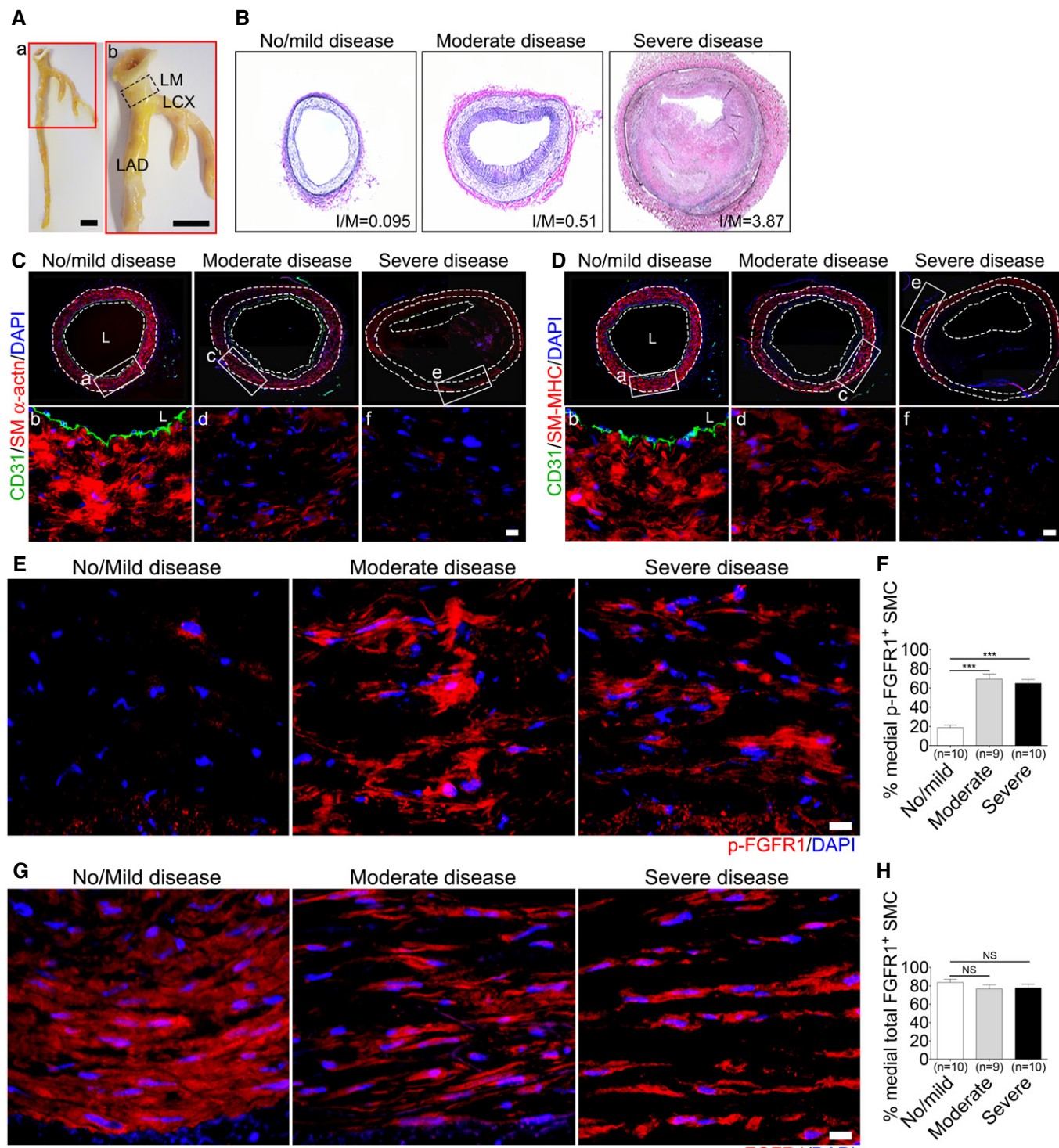

**Figure 4.**

**Table 1.  Human subject characteristics[a].**

| | Disease severity by I/M ratio | | | |
| --- | --- | --- | --- | --- |
| | No/Mild I/M < 0.2 0.14 ± 0.03 *n* = 10 | Moderate I/M 0.2–1.0 0.4 ± 0.2 *n* = 15 | Severe I/M > 1.0 2.0 ± 1.4 *n* = 18 | *P*-value |
| Explanted hearts | | | | |
| Organ donors | 6 (60.0) | 8 (53.3) | 10 (55.6) | 0.9470 |
| Transplant recipients | 4 (40.0) | 7 (46.7) | 8 (44.4) | 0.9470 |
| Demographics | | | | |
| Age (years) | 42.3 ± 13.9 | 56.7 ± 8.3 | 61.6 ± 6.5 | <0.0001 |
| Male | 4 (40.0) | 10 (66.7) | 13 (72.2) | 0.2226 |
| Caucasian | 7 (70.0) | 10 (66.7) | 15 (83.3) | 0.5149 |
| Past medical history | | | | |
| Coronary artery disease | 0 (0.0) | 0 (0.0) | 9 (50.0) | 0.0004 |
| Cerebrovascular disease | 1 (10.0) | 0 (0.0) | 4 (22.2) | 0.1377 |
| Peripheral vascular disease | 1 (10.0) | 1 (6.7) | 2 (11.1) | 0.9053 |
| Atherosclerosis risk factors | | | | |
| Diabetes mellitus | 2 (20.0) | 3 (20.0) | 6 (33.3) | 0.6135 |
| Hypertension | 4 (40.0) | 9 (60.0) | 11 (61.1) | 0.5155 |
| Hyperlipidemia | 2 (20.0) | 3 (20.0) | 7 (46.7) | 0.3954 |
| Tobacco use | 4 (40.0) | 7 (46.7) | 10 (55.6) | 0.7168 |
| Obesity | 4 (40.0) | 2 (13.3) | 5 (27.8) | 0.3135 |

[a]Left main coronary arteries were procured from the explanted hearts of 43 individuals within the operating room at either organ donation or cardiac transplantation. The degree of atherosclerotic disease was quantified as intima-to-media (I/M) ratio and de-identified clinical data were recorded. Data represent number (%) or means ± SD. Comparisons between groups of categorical variables were by chi-square test and of continuous variables were by one-way ANOVA.

cholesterol, triglyceride, HDL-C levels, aorta diameter, or heart function (Fig EV4).

Aortas from $Frs2\alpha^{SMCKO}/Apoe^{-/-}$ and $Apoe^{-/-}$ mice were examined after eight (Fig EV5A and B) or sixteen (Fig 7A and B) weeks of high-fat diet. In both cases, $Frs2\alpha^{SMCKO}/Apoe^{-/-}$ animals demonstrated much lower extent of the total aorta atherosclerotic plaque burden. Notably, the progression of atherosclerosis was markedly reduced in $Frs2\alpha^{SMCKO}/Apoe^{-/-}$ mice compared to $Apoe^{-/-}$ controls: By 8 weeks, there was a 43% decrease in the total aorta plaque size (5.57% in $Apoe^{-/-}$ versus 3.16% in $Frs2\alpha^{SMCKO}/Apoe^{-/-}$) (Fig EV5B) and by 16 weeks 50% decrease (17.04% in $Apoe^{-/-}$ versus 8.47% in $Frs2\alpha^{SMCKO}/Apoe^{-/-}$) (Fig 7B).

Histochemical analysis of plaques showed a ~50% reduction in plaque cellularity (335 cells/plaque in $Apoe^{-/-}$ versus 164 cells/plaque in $Frs2\alpha^{SMCKO}/Apoe^{-/-}$) (Fig 7C and F). Furthermore, Movat staining demonstrated that fibrous caps were thicker and necrotic core was smaller in $Frs2\alpha^{SMCKO}/Apoe^{-/-}$ compared to $Apoe^{-/-}$ mice (Fig 7D and G). Finally, Ki67 staining demonstrated reduced proliferation rate in plaque as well as media cells (Fig 7E and H). All of these findings are consistent with a more stable plaque phenotype. Consistent with these changes in plaque cellularity and fibrous cap size, there was a decrease in the plaque SM α-actin area (12.82 in $Apoe^{-/-}$ versus 7.28 in $Frs2\alpha^{SMCKO}/Apoe^{-/-}$) and increased collagen deposition (0.83 in $Apoe^{-/-}$ versus 1.56 in $Frs2\alpha^{SMCKO}/Apoe^{-/-}$) (Fig EV5C and D).

## Discussion

The results of this study demonstrate the importance of SMC proliferation in atherosclerosis and implicate FGF regulation of TGFβ signaling as an important controller of this process. Several lines of evidence support these conclusions. In *in vitro* SMC culture assays, FRS2α knockdown markedly increased TGFβ signaling leading to induction of a contractile phenotype and suppression of cell proliferation even in the presence of serum. *In vivo*, SMC FRS2α deletion decreased the extent of atherosclerosis in $Apoe^{-/-}$ mice. Not only was there a reduction in the size of atherosclerotic plaque and decreased plaque cellularity, but the plaque morphology was also altered with a decrease in the size of necrotic core and increased fibrous cap, findings consistent with a "stable" plaque. Finally, in human clinical specimens, there was a strong correlation between the activation of FGF signaling, loss of TGFβ activity, and the extent of atherosclerotic coronary artery disease. Collectively, these results demonstrate that FGF activity plays a key role in SMC biology by inhibiting TGFβ signaling, thereby leading to the loss of a differentiated, contractile phenotype and conferring a capacity for SMC proliferation, migration, and neointima formation. In contrast, inhibition of SMC FGF signaling promotes reversion of active proliferating cells to the contractile phenotype and limits the extent of atherosclerosis.

The switch from the differentiated contractile SMC phenotype to the less differentiated, proliferative state has long being recognized

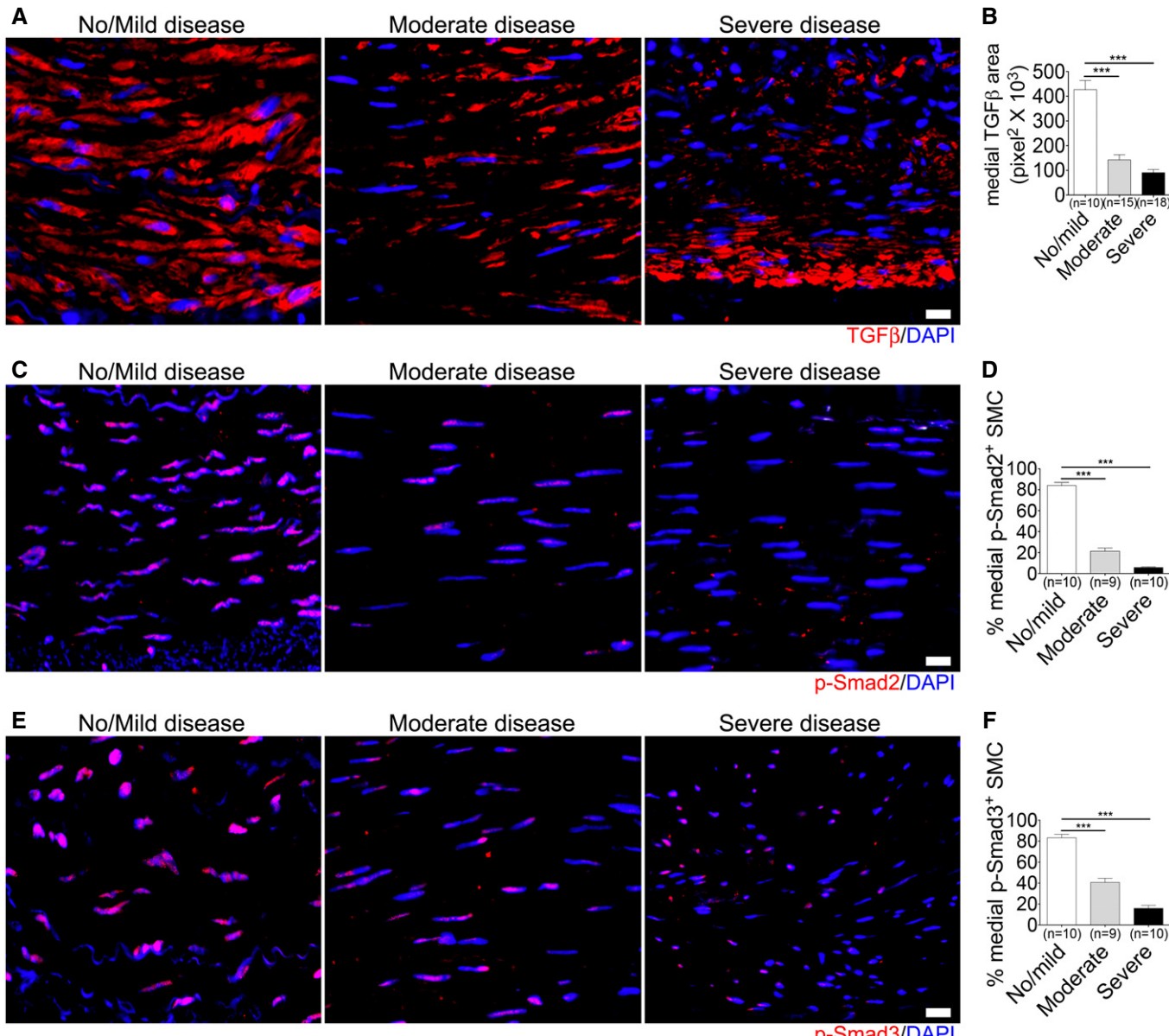

**Figure 5.  TGFβ signaling activity in smooth muscle cells in human left main coronary arteries with various degrees of atherosclerosis.**

A, B    Representative images of immunofluorescence staining for TGFβ (red) from patients with no/mild, moderate, or severe disease. Nuclei were counterstained with DAPI (blue). Scale bar: 16 μm. Medial TGFβ area is shown in (B) (***$P < 0.001$ compared to no/mild disease; one-way ANOVA with Newman–Keuls *post hoc* test for multiple comparison correction).

C, D    Representative images of immunofluorescence staining for p-Smad2 (red) from patients with no/mild, moderate, or severe disease. Nuclei were counterstained with DAPI (blue). Scale bar: 16 μm (***$P < 0.001$ compared to no/mild disease; one-way ANOVA with Newman–Keuls *post hoc* test for multiple comparison correction).

E, F    Representative images of immunofluorescence staining for p-Smad3 (red) from patients with no/mild, moderate, or severe disease. Nuclei were counterstained with DAPI (blue). Scale bar: 16 μm (***$P < 0.001$ compared to no/mild disease; one-way ANOVA with Newman–Keuls *post hoc* test for multiple comparison correction).

Data information: The data shown in (D, F) are the means ± SD of the percentage of medial p-Smad2 or p-Smad3-positive SMCs. Images are representative of 10 No/mild, 9 moderate, and 10 severe disease human left main coronary artery samples. A full table of *P*-values for this figure is shown in Appendix Table S1.

as a key event in a number of illnesses where SMC proliferation plays an important role. Atherosclerosis, a disease characterized by lipid deposition-induced inflammation and macrophage accumulation in the arterial wall (Tabas *et al*, 2015), is one such example. A SMC phenotype switch, characterized by decreased expression of differentiation markers and increased proliferation, migration, and

SMC-dependent production of ECM proteins, has been previously observed in atherosclerotic lesions (Doran *et al*, 2008; Gomez & Owens, 2012).

FGFs are extensively produced in atherosclerotic lesions by proliferating SMCs themselves as well as by a number of other cell types (Lindner & Reidy, 1991; Casscells *et al*, 1992) and

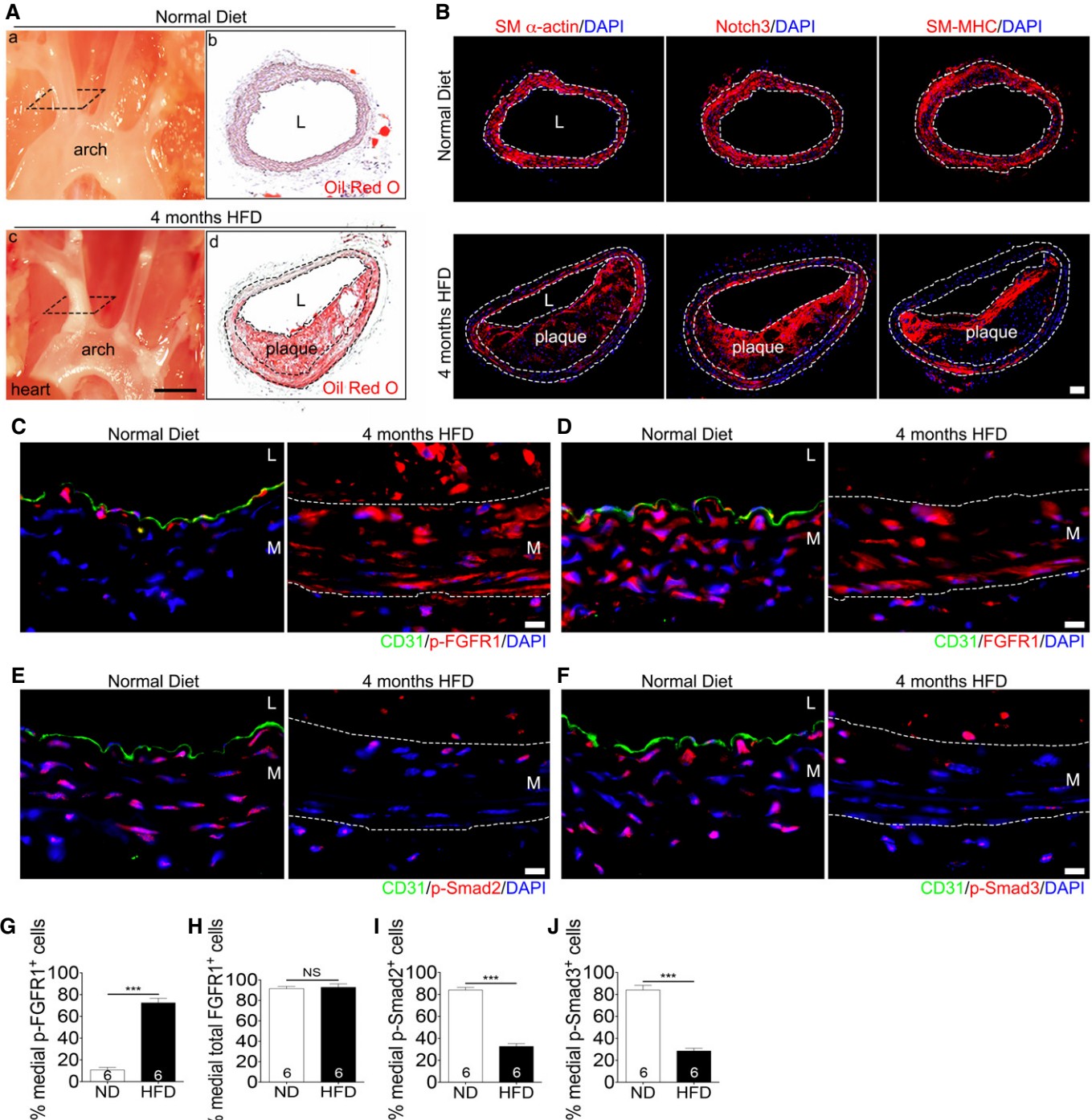

**Figure 6. FGFR1 and TGFβ signaling activity in smooth muscle cells in a mouse atherosclerosis model.**

A   Dissected mouse aorta demonstrating lipid-rich plaques in brachiocephalic artery after 4 months of high-fat diet compared to the normal diet in *Apoe*[−/−] mice. Panels b, d are cross sections of brachiocephalic artery from a, c stained with Oil Red O. L: lumen. Scale bar: 4 mm. *n* = 3 mice per group.

B   Histological analysis of mouse normal artery or atherosclerotic plaque in brachiocephalic artery with anti-SM α-actin, anti-Notch3, and anti-SM-MHC antibodies. Nuclei were counterstained with DAPI (blue). Scale bar: 62 μm. *n* = 3 mice per group.

C–F   Analysis of brachiocephalic artery of *Apoe*[−/−] mice maintained for 4 months on either normal or high-fat diet using anti-CD31 (green), anti-p-FGFR1 (red; C), anti-FGFR1 (red; D), anti-p-Smad2 (red; E), and anti-p-Smad3 (red; F) antibodies. Nuclei were counterstained with DAPI (blue). Scale bar: 62 μm. L: lumen. M: media. *n* = 6 mice per group.

G–J   Quantification of the number of medial smooth muscle cells expressing p-FGFR1, FGFR1, p-Smad2, and p-Smad3. ND: normal diet. HFD: high-fat diet.

Data information: All data shown as means ± SD (***$P < 0.001$ compared to ND, NS: not significant compared to ND; unpaired two-tailed Student's *t*-test). A full table of *P*-values for this figure is shown in Appendix Table S1.

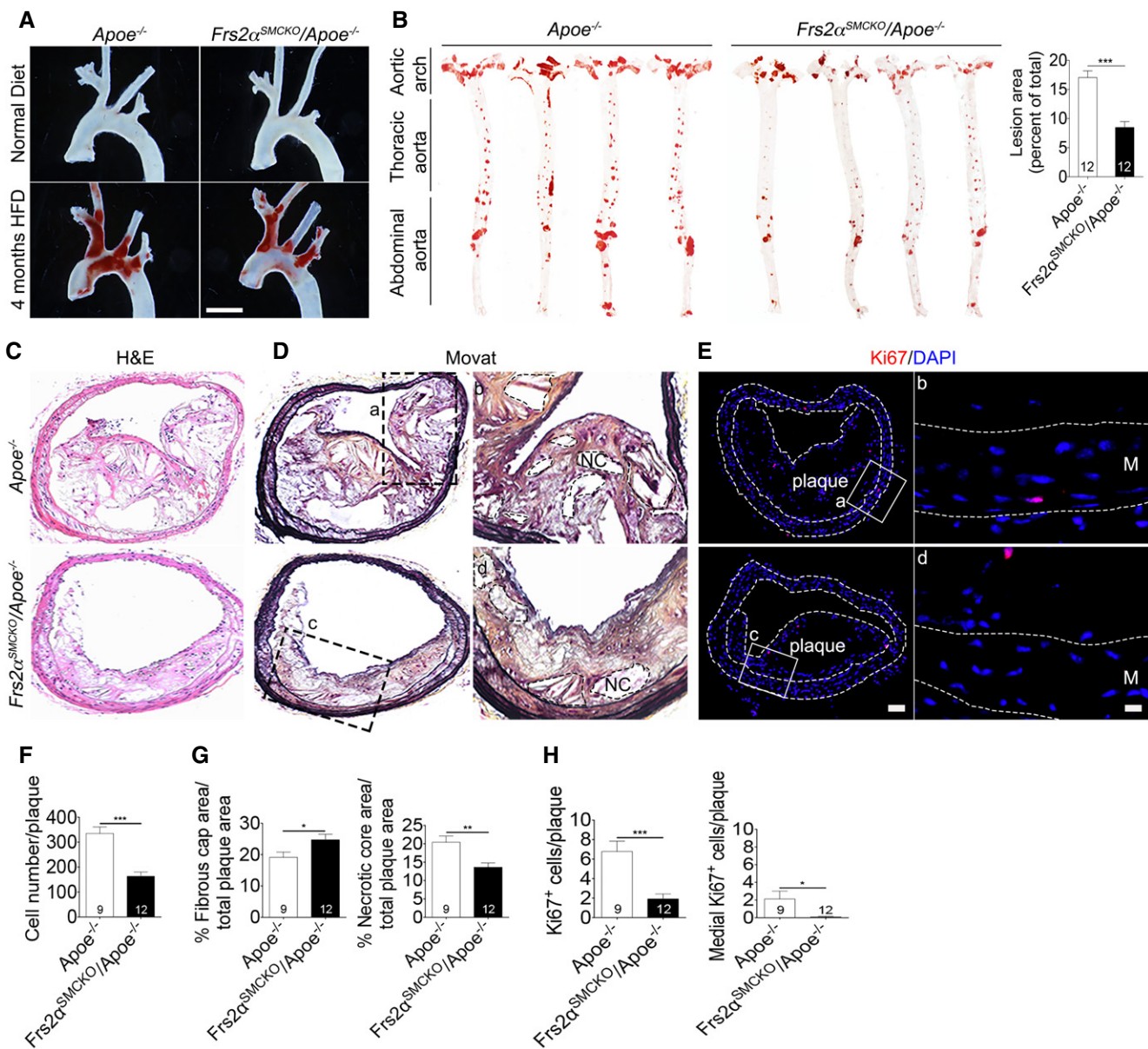

**Figure 7. Smooth muscle cell FRS2α knockout inhibits atherosclerosis plaque development after 16 weeks of high-fat diet.**

A    Representative photomicrographs of Oil Red O-stained atherosclerotic lesions in the aortic arch of *Apoe*⁻/⁻ or *Frs2*^SMCKO^/*Apoe*⁻/⁻ mice after 16 weeks of high-fat diet. *n* = 3 mice per group. Scale bar: 5 mm.

B    (Left) Microphotographs of aortas (en face) from *Apoe*⁻/⁻ and *Frs2*^SMCKO^/*Apoe*⁻/⁻ mice after 16 weeks of high-fat diet after staining with Oil Red O. (Right) Lesion area quantification (***$P < 0.001$ compared to *Apoe*⁻/⁻; unpaired two-tailed Student's *t*-test; *n* = 12 mice per group).

C, D  Representative cross sections of brachiocephalic arteries of *Apoe*⁻/⁻ and *Frs2*^SMCKO^/*Apoe*⁻/⁻ mice stained with hematoxylin and eosin (H&E) (C) and Movat (D). Panels b, d are high-magnification view from a, c. NC: necrotic core. *Apoe*⁻/⁻ mice *N* = 9, *Frs2*^SMCKO^/*Apoe*⁻/⁻ mice *N* = 12.

E    Histological analysis of atheosclerotic plaque with anti-Ki67 antibody. Nuclei were counterstained with DAPI (blue). Panels b, d are high-magnification images from a, c. Scale bar: 62 μm (low-magnification images); 10 μm (high-magnification images). *Apoe*⁻/⁻ mice *N* = 9, *Frs2*^SMCKO^/*Apoe*⁻/⁻ mice *N* = 12.

F    Quantification of plaque cellularity.

G    Quantifications of the extent of fibrous cap and necrotic areas in brachiocephalic artery of A*poe*⁻/⁻ and *Frs2*^SMCKO^/*Apoe*⁻/⁻ mice.

H    Measurement of Ki67⁺ cells.

Data information: All data shown as mean ± SD (*$P < 0.05$, **$P < 0.01$, ***$P < 0.001$ compared to *Apoe*⁻/⁻; unpaired two-tailed Student's *t*-test). A full table of *P*-values for this figure is shown in Appendix Table S1.

their ability to stimulate SMC proliferation is well established. Indeed, both an antibody against FGF2 and an FGFR tyrosine kinase inhibitor (SU5402) have been shown to inhibit SMC proliferation in the setting of vascular injury or atherosclerosis (Lindner & Reidy, 1991; Raj *et al*, 2006). Our data are consistent with these observations and implicate FGF regulation of TGFβ

signaling as the critical molecular pathway controlling these events.

While it is clear that FGF signaling is central to control of TGFβ and SMC differentiation, the exact FGF(s) involved is uncertain. FGFs are a family of 22 secreted proteins that regulate cell migration, proliferation, and differentiation, among other biologic processes (Eswarakumar et al, 2005; Thisse & Thisse, 2005). FGFs mediate their cellular responses by binding to and activating a family of 4 receptor tyrosine kinases (FGFR1-4) which display different ligand-binding characteristics and biologic functions. A number of FGF family members are present in the serum as well as in the atherosclerotic plaques themselves (Brogi et al, 1993; Hughes, 1996) and a variety of cell types have the ability to secrete FGFs and activate FGFR1, the principle FGFRs in SMCs. In particular, macrophages, T, and B lymphocytes are an abundant source of FGFs as are damaged and dying SMCs (Murakami & Simons, 2008). Thus, atherosclerotic plaques are a particularly FGF-rich environment. Indeed, our observations of increased FGFR1 activity, reduced p-Smad2 and p-Smad3

levels, and the loss of contractile markers in both human and mouse atherosclerotic lesions are consistent with increased FGF signaling.

To establish a causal relationship between SMC FGF signaling and atherosclerotic plaque growth, we targeted expression of $Frs2\alpha$, an adaptor protein expressed in all SMCs (Chen et al, 2009) and involved in activation of MAPK by all FGFRs (Gotoh, 2008). Since a global embryonic $Frs2\alpha$ gene knockout is lethal (Hadari et al, 2001), we generated a SMC-specific knockout using constitutively active SM22αCre. The analysis of $Frs2\alpha^{SMCKO}$ mice showed that FGF signaling is not required for the development and basal homeostatic functions of SMCs, suggesting that its function can be compensated by other signaling pathways. However, $Frs2\alpha^{SMCKO}/Apoe^{-/-}$ mice had much smaller atherosclerotic plaques than their $Apoe^{-/-}$ counterparts. Importantly, in addition to being smaller, the plaques in $Frs2\alpha^{SMCKO}/Apoe^{-/-}$ mice also had a reduction in the necrotic core area, a feature associated with increased plaque stability. The latter finding is particularly significant as the absence of SMC proliferation was thought to lead to larger size necrotic core and a more unstable plaque (Weissberg et al, 1996).

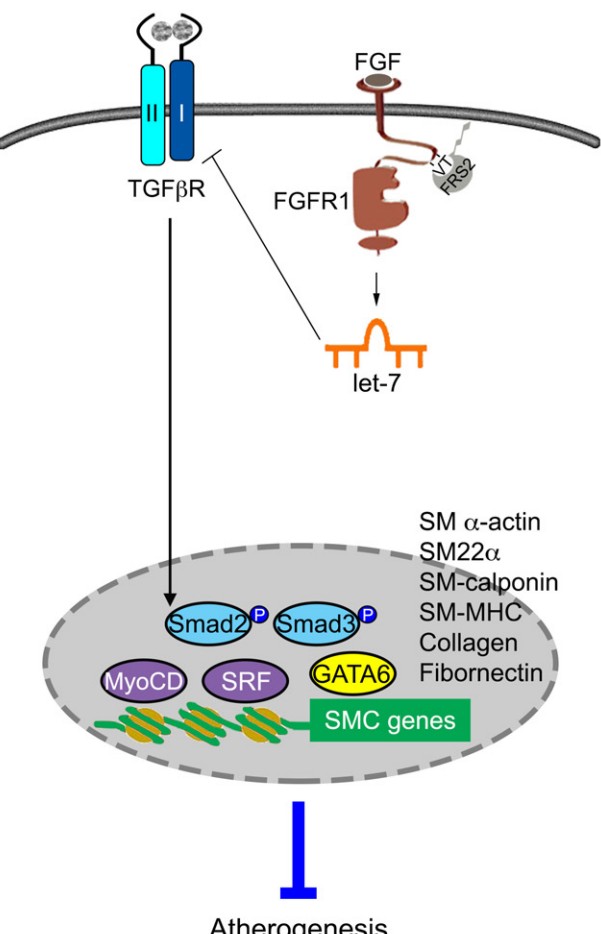
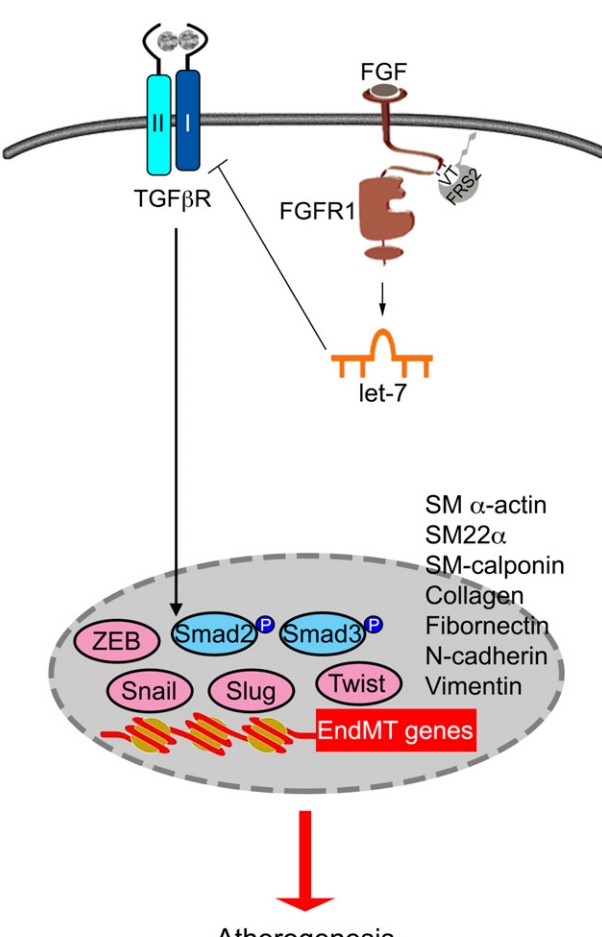

**Figure 8. Scheme of FGF-dependent regulation of TGFβ signaling in smooth muscle cells and endothelial cells.**
In both smooth muscle cells and endothelial cells, suppression of FGF signaling leads to reduction in let-7 miRNA expression that, in turn, results in increased TGFβR1 expression and activation of TGFβ-dependent transcriptional program. In SMC (left panel), activation of TGFβ signaling promotes SMC conversion from proliferative-to-contractile phenotype, thereby reducing the number of SMCs in the plaque and reducing plaque growth. In contrast, EC activation of TGFβ signaling promotes endothelial-to-mesenchymal transition, thus increasing the number of plaque SMCs and promoting plaque growth.

The origin of neointima SMCs in atherosclerotic plaque and contribution of medial SMC proliferation to the plaque growth are hotly debated and controversial issues. A number of potential sources for the origin of plaque SMCs have been proposed, including dedifferentiated medial SMCs, resident progenitor cells, adventitial fibroblasts and macrophages, and endothelial-to-mesenchymal transition (EndMT) (Hu *et al*, 2002; Bentzon *et al*, 2006; Hoofnagle *et al*, 2006; Tanaka *et al*, 2008; Chen *et al*, 2015; Nurnberg *et al*, 2015; Shankman *et al*, 2015). The decrease in neointima size observed in $Frs2\alpha^{SMCKO}/Apoe^{-/-}$ mice may have arisen from several factors. Most likely is the reduction in medial SMC proliferation although decreased extracellular matrix production by contractile (versus proliferative) SMCs may have also played a role.

Nevertheless, the continued presence of SMCs in the atherosclerotic plaque and persistent, albeit much reduced plaque growth, despite inhibition of SMC proliferation suggest that other sources may still contribute. One recently described source of SMCs is the phenomenon of endothelial-to-mesenchymal transition that can, under certain conditions, lead to significant accumulation of endothelium-derived SMCs in the neointima (Chen *et al*, 2012, 2014, 2015; Maddaluno *et al*, 2013; Cooley *et al*, 2014). Interestingly, EndMT is induced by the same FGF/TGFβ antagonism that in SMCs inhibits SMC proliferation and leads to reduction in atherosclerosis. In both cell types, the suppression of FGF signaling leads to reduction in *let-7* miRNA expression that results in increased TGFβR1 expression and activation of TGFβ signaling and activation of SMC transcriptional program (Fig 8). In SMC, this leads to a proliferative-to-contractile phenotype shift and arrest of SMC proliferation, thereby reducing atherosclerotic plaque cellularity and growth. In contrast, in ECs this leads to increased production of SMC and increased plaque growth (Fig 8). These considerations suggest that inhibition of endothelial and activation of smooth muscle cell TGFβ signaling would be effective as atherosclerosis treatment while systemic TGFβ in inhibition would be ineffective.

In summary, this study demonstrates that TGFβ-driven induction of SMC proliferative-to-contractile phenotype, achieved by suppression of FGF SMC signaling input, reduces growth of atherosclerotic plaques. The FGF-dependent regulation of TGFβ activity appears to play a significant role in the development of atherosclerotic lesions and may thus represent an important new therapeutic target.

# Materials and Methods

### Chemicals

The TGFβR1 kinase inhibitor SB431542 (Sigma S4317) was reconstituted in DMSO (Sigma D2650) and used at a final concentration of 10 μM in cell culture.

### Antibodies

We used the following antibodies for immunoblotting (IB), immunofluorescence (IF), or immunohistochemistry (IHC): BrdU Alexa Fluor 594 (Life Technologies B35132; IF 1:100), calponin (Sigma C2687; IB 1:1,000, IF 1:500), CD31 (Santa Cruz sc-1506; IHC 1:100 for mouse paraffin samples), CD31 (BD 561814; IHC 1:100 for mouse fixed optimal cutting temperature (OCT) samples), CD31 (Dako M0823; IHC 1:200 for human frozen samples), collagen 1 (Novus Biologicals NB600-408; IHC 1:200), cyclin D1 (Santa Cruz sc-20044; IB 1:1,000), FGFR1 (Epitomics 2144-1; IB 1:500), FGFR1 (phospho-Y654) (Abcam ab59194; IHC 1:100), FGFR1 (Abcam ab10646; IHC 1:100), FRS2α (Abcam ab10425; IHC 1:100), FRS2α (Santa Cruz sc-8318; IB 1:1,000), GAPDH (glyceraldehyde phosphate dehydrogenase) (Cell Signaling #2118; IB 1:1,000), HSP90 (Sigma 4300541; IB 1:1,000), Ki67 (Cell Signaling #9027; IHC 1:100), myosin (smooth) (Sigma M7786; IB 1:1,000), Notch3 (ab23426, Abcam; IHC 1:100), p21 (Cell Signaling #2947; IB 1:1,000), p27 (Cell Signaling #3688; IB 1:1,000), SM22α (Abcam ab14106; IB 1:2,000, IF 1:1,000), phospho-Smad2 (Ser$^{465/467}$) (Cell Signaling #3108; IB 1:1,000), phospho-Smad2 (Ser$^{465/467}$) (Cell Signaling #3101; IHC for human paraffin samples 1:100) (Millipore AB3849; IHC for mouse paraffin samples), phospho-Smad3 (Ser$^{465/467}$) (R&D AB3226; IB 1:1,000), phospho-Smad3 (Ser$^{465/467}$) (Abcam ab52903; IHC 1:100), Smad2 (Cell Signaling #3122; IB 1:1,000), Smad2/3 (BD 610843; IB 1:1,000), smooth muscle α-actin (Sigma A2547; IB 1:2,000, IHC 1:400), smooth muscle α-actin-Cy3 (Sigma C6198; IF 1:1,000), smooth muscle α-actin-APC (allophycocyanin) (R&D IC1420A; IHC 1:50), smooth muscle myosin heavy chain 11 (SM-MHC 11) (Abcam ab683; IHC 1:100), TGFβ (Abcam ab66043; IHC 1:100), TGFβR1 (Santa Cruz sc-398; IB 1:1,000), TGFβR2 (Santa Cruz sc-400; IB 1:1,000), and β-tubulin (Sigma T7816; IB 1:2,000).

### Cell culture and reagents

Human 293T T17 cells (human embryonic kidney cells, ATCC CRL-11268) were maintained in Dulbecco's modified Eagle's medium (Gibco 11965-092) with 10% fetal bovine serum (Life Technologies 16000-044) and penicillin–streptomycin (15140-122, Gibco), and were grown at 37°C, 5% CO$_2$. Human aortic smooth muscle cells (#C-007-5C), media (#M231-500), and supplements (SMGS: S-007-25; SMDS: S-008-5) were purchased from Life Technologies. The cells were grown at 37°C, 5% CO$_2$ in Medium 231 supplemented with smooth muscle growth supplement (SMGS containing 4.9% FBS, 2 ng/ml FGF2, 0.5 ng/ml EGF, 5 ng/ml heparin, 2 μg/ml IGF-1, and 0.2 μg/ml BSA). For SMC differentiation, HASMCs were incubated with Medium 231 containing smooth muscle differentiation supplement (SMDS containing 1% FBS and 30 μg/ml heparin) for different time points. Primary human aortic smooth muscle cells between passages 6 and 10 were used in all experiments.

### Generation of lentiviruses

Human FGFR1, human Smad2, and human TGFβR2 shRNA lentiviral constructs were purchased from Sigma and human FRS2α shRNA lentiviral construct was purchased from Open Biosystems. For the production of shRNA lentivirus, 3.7 μg of Δ8.2, 0.2 μg of VSVG, and 2.1 μg of pLKO.1 carrying the control, FGFR1, FRS2α, Smad2, or TGFβR2 shRNA were co-transfected into 293T cells using X-tremeGENE 9 DNA Transfection Reagent (Sigma 6365787001). Forty-eight hr later, the medium was harvested, cleared by 0.45-μm filter (PALL Life Sciences 4184), mixed with polybrene (5 μg/ml) (Sigma H9268), and applied to cells. After 6-h incubation, the virus-containing medium was replaced by the fresh medium.

For production of *let-7* miRNA lentivirus, 10 µg of pMIRNA1 carrying the *let-7*b (PMIRHlet7bPA-1) miRNA expression cassette (System Biosciences), 5 µg of pMDLg/PRRE, 2.5 µg of RSV-REV, and 3 µg of pMD.2G were co-transfected into 293T cells using X-tremeGENE 9 DNA transfection reagent (Sigma 6365787001). Forty-eight hr later, the medium was harvested, cleared by 0.45-µm filter (PALL Life Sciences 4184), mixed with 5 µg/ml polybrene (Sigma H9268), and applied to cells. After 6-h incubation, the virus-containing medium was replaced by fresh medium.

### RNA isolation and qRT–PCR

Cells were suspended in TRIzol reagent (Invitrogen #15596018), and total RNA (QIAGEN #74134)- and miRNA-enriched fraction (QIAGEN #74204) were isolated according to the manufacturer's instructions. Reverse transcriptions were performed by using iScript cDNA synthesis kit (Bio-Rad 170-8891) for mRNA or $RT^2$ miRNA First Strand kit (QIAGEN 331401) for miRNA. qRT–PCR was performed using Bio-Rad CFX94 (Bio-Rad) by mixing equal amount of cDNAs, iQ SYBR Green Supermix (Bio-Rad 170-8882) and gene-specific primers SABiosciences (a QIAGEN company) (ACTB [PPH00073G], Actb [PPM02945B], COL1a1 [PPH01299F], COL3a1 [PPH00439E], CTGF [PPH00550G], ELN [PPH06895F], FGFR1 [PPH00372F], FGFR2 [PPH00391E], FGFR3 [PPH000382A], FGFR4 [PPH00390B], FRS2 [PPH01645E], Frs2 [PPM04336A], GATA6 [PPH06943F], KL [PPH13489A], KLB [PPH10455A], MKL1 [PPH01263A], MKL2 [PPH12812A], MYOCD [PPH05713A], NOTCH3 [PPH06020B], CDKN1A [PPH00211E], CDKN1B [PPH00212C], SERPINE1 [PPH00215F], ACTA2 [PPH01300B], TAGLN [PPH19531F], CNN1 [PPH02065A], SRF [PPH00707A], TGFB1 [PPH00508A], TGFB2 [PPH00524B], TGFB3 [PPH00531E], TGFBR1 [PPH00237C], TGFBR2 [PPH00339C], let-7a [MPH00001A], let-7b [MPH00002A], let-7c [MPH00003A], let-7d [MPH00004A], let-7e [MPH00005A], let-7f [MPH00006A], let-7 g [MPH00007A], let-7i [MPH00008A], miR-98 [MPH00480A], and SNORD47 [MPH01660A]). All reactions were done in a 25 µl reaction volume in duplicate. Individual mRNA or miRNA expression was normalized in relation to expression of endogenous β-actin or small nuclear SNORD47, respectively. PCR amplification consisted of 10 min of an initial denaturation step at 95°C, followed by 46 cycles of PCR at 95°C for 15 s, 60°C for 30 s (for mRNA cDNA), and 10 min of an initial denaturation step at 95°C, followed by 46 cycles of PCR at 95°C for 15 s, 55°C for 30 s, and 70°C for 30 s (for miRNA cDNA).

### Western blot analysis

Cells were lysed with HNTG lysis buffer (20 mM HEPES, pH 7.4/150 mM NaCl/10% glycerol/1% Triton X-100/1.5 mM $MgCl_2$/1.0 mM EGTA) containing complete mini EDTA-free protease inhibitors (Sigma #11836170001) and phosphatase inhibitors (Sigma #04906837001). 20 µg of total protein from each sample was resolved on Criterion TGX Precast Gels (Bio-Rad #567-1084) with Tris/glycine/SDS running buffer (Bio-Rad #161-0772), transferred to nitrocellulose membranes (Bio-Rad #162-0094) and then probed with various antibodies. Chemiluminescence measurements were performed using SuperSignal West Pico Chemiluminescent Substrate (Thermo Fisher Scientific Prod #34080).

### Quantification of Western blots

Images of blot signals on HyBlot ES® Autoradiography Film (DENVILLE E3218) were scanned on a CanoScan LiDE 200 scanner. Images were then viewed in ImageJ software for data analysis. Signal intensities of individual bands were determined using gel analysis followed the ImageJ user's guide. Data were exported to GraphPad Prism Software to generate the plot. To obtain the mean, standard deviation, and test for significant differences between samples, we averaged the relative band intensities from three to four independent experiments. Data are presented as fold change in protein expression for the experimental groups compared to the control group after normalized to loading controls (GAPDH, HSP90, β-tubulin, or total phosphoprotein). Error bars showed the calculated standard deviation. Statistical significance was calculated by two-tailed Student's *t*-test. *P*-values of $< 0.05$ were considered significant and are indicated with asterisks.

### Immunofluorescence staining

Cultured primary human aortic smooth muscle cells were grown on 10 µg/ml fibronectin (Sigma F2006)-coated glass-bottomed dishes (MatTek CORPORATION P35G-1.5-20-C). Cells were first fixed with 2% paraformaldehyde (Polysciences, Inc, 18814) in PBS for 20 min at 37°C, then permeabilized with 0.1% Triton X-100 in PBS containing 2% PFA at room temperature for 5 min, and blocked with 3% bovine serum albumin (Jackson ImmunoResearch Laboratories, Inc. 001-000-162) at room temperature for 60 min. Cells were washed with PBS and incubated with SM α-actin-Cy3 (1:1,000 in 1% BSA), SM22α (1:1,000 in 1% BSA), and SM-calponin (1:500 in 1% BSA) antibodies at 4°C overnight, washed three times with PBS, and incubated with diluted Alexa Fluor-conjugated secondary antibody (1:500) (Life Technologies) for 1 h at room temperature. The dishes were then washed three times with PBS and mounted using Prolong Gold antifade reagent with DAPI (Life Technologies P36931).

### Cell contraction assay

Cell contraction assay was evaluated using a Cell Contraction Assay kit according to the manufacturer's instructions (CELL BIOLABS CBA-201). Briefly, HASMCs were harvested and suspended at $5 \times 10^5$ cells/ml, and the collagen lattice was prepared by mixing two parts of cell suspension and eight parts of cold collagen gel solution. Subsequently, 500 µl of the cell–collagen mixture was cast into each well of a 24-well plate and allowed to polymerize at 37°C for 1 h. After collagen polymerization, cells were incubated in SMC growth medium (Medium 231 plus SMGS) for 24 h. During which stress developed. Upon release of the collagen lattice from the culture dish, the embedded cells become free to contract the deformable lattice, thus reducing its surface area. This was quantified 24 h after detachment of the gel from the dish using ImageJ and expressed as the percentage of the area of the entire well.

### xCELLigence real-time cell analysis (RTCA)

Cell proliferation experiments were carried out using the *xCELLigence* RTCA DP instrument (Roche Diagnostics GmbH) in a humidified

incubator at 37°C and 5% $CO_2$. Cell proliferation experiments were performed using modified 16-well plates (E-plate, Roche Diagnostics GmbH). Initially, 100 μl of cell-free growth medium was added to the wells. After leaving the devices at room temperature for 30 min, the background impedance for each well was measured. 100 μl of the cell suspension was then seeded into the wells (1,000 cells/well). Plates were locked in the RTCA DP device in the incubator and the impedance value of each well was automatically monitored by the *xCELLigence* system and expressed as a *cell index* value (CI). CI was monitored every 15 min for 600 times. Two replicates of each cell concentration were used in each test. All data have been recorded by the supplied RTCA software (version 1.2.1).

### WST-1 cell proliferation assay

Cell proliferation was assayed using a WST-1 Cell Proliferation Assay System (Sigma 05 015 944 001) according to the manufacturer's instructions. Briefly, control, FRS2α shRNA-, TGFβR2 shRNA-, Smad2 shRNA-, or SB431542-treated HASMCs were plated at a density of 5,000 cells per well in a 48-well plate in 200 μl culture medium. To evaluate cell proliferation, cells were incubated for 1–3 days and subsequently exposed to 20 μl WST-1 reagent for 1 h at 37°C in 5% $CO_2$. The absorbance of the treated samples against a blank control was measured at 450 nm as the detection wavelength and 670 nm as the reference wavelength for the assay.

### Cell cycle analysis

Cell cycle analysis was performed using propidium iodide (PI) staining and flow cytometry. Cells were trypsinized, washed twice in PBS, and fixed in 70% ethanol at −20°C overnight. After washing twice in PBS, the cells were treated with 100 μg/ml RNase A (Sigma R5125) at 37°C for 30 min and stained in 50 μg/ml PI solution (Sigma P4170). Then, the cells were transferred to flow cytometry tubes with filters (BD #352235) for cell cycle analysis. Ten thousand events were collected for each sample. The data were collected and analyzed with FlowJo software (Tree Star).

### *In vitro* BrdU labeling and detection

Cultured primary human aortic smooth muscle cells were grown on 10 μg/ml fibronectin (Sigma F2006)-coated glass-bottomed dishes (MatTek CORPORATION P35G-1.5-20-C). At the end of the experimental protocol, HASMCs were labeled with 10 μM BrdU (5-bromo-2′-deoxyuridine) (Life Technologies B23151) for 2 h at 37°C. After labeling, cells were fixed with 4% paraformaldehyde (Polysciences, Inc, 18814) in PBS for 15 min at room temperature, then permeabilized with 0.1% Triton X-100 in PBS at room temperature for 20 min. The cells were then incubated in 1N HCl on ice for 10 min, 2N HCl at room temperature for 10 min followed by phosphate/citric acid buffer (pH 7.4) at room temperature for 10 min. Cells were incubated with BrdU Alexa Fluor 594 (Life Technologies B35132) (1:100 in PBS/0.1% Triton/5% normal goat serum) at room temperature for 2 h. The dishes were then washed three times with 0.1% Triton X-100 in PBS and mounted

using Prolong Gold antifade reagent with DAPI (Life Technologies P36931).

### BrdU ELISA assay

To quantify the degree of HASMCs proliferation, BrdU assays were performed using the Cell Proliferation ELISA BrdU Colorimetric Assay kit (Sigma 11647229001). HASMCs were plated in 96-well plates at a density of 1,000 cells per well. At the end of the experimental protocol, HASMCs were labeled with BrdU (10 μM) for 2 h at 37°C. The HASMCs were then fixed and denatured for 30 min followed by exposure to a peroxidase-conjugated anti-BrdU antibody for 90 min all at room temperature. The HASMCs were then washed with PBS three times followed by incubation with a peroxidase substrate solution at room temperature until the development of a noticeable color sufficient for photometric detection in which at that point the reaction was stopped using 1 M $H_2SO_4$. The degree of color change was quantified using the BioTek Synergy 2 Multimode microplate reader to determine the degree of cell proliferation. An absorbance wavelength of 450 nm and reference wavelength of 690 nm were used.

### Patient population

Human coronary arteries were obtained from the explanted hearts of transplant recipients or cadaver organ donors. Research protocols were approved by the Institutional Review Boards of Yale University and the New England Organ Bank. A waiver for consent was approved for surgical patients and written informed consent was obtained from a member of the family for deceased organ donors. Table 1 summarizes clinical characteristics of this patient group.

### Specimen collection

Investigators were on call with the surgical team and collected the heart at the time of explant. To minimize *ex vivo* artifacts, a ~5–20 mm segment of the left main coronary artery was removed within the operating room (Fig 4A) and immediately processed as frozen sections in optimal cutting temperature medium and, when of sufficient length, an additional segment was also fixed in formalin for later embedding, sectioning, and staining.

### Generation of mice

Mice were all bred on a C57BL/6 background. *Frs2α*^flox/flox^ mice were previously described (Lin *et al*, 2007). *Frs2α*^flox/flox^ mice were bred with mice expressing Cre recombinase under the *SM22α* promoter. *SM22α*-Cre;*Frs2α*^flox/flox^ offspring were crossed to C57BL/6 *Apoe*^−/−^ mice (JAX SN:002052). PCR genotyping analysis was done using the following primers: *Frs2*^flox/flox^ (5′-GAGTGTGCTGT-GATTGGAAGGCAG-3′ and 5′-GGCACGAGTGTCTGCAGACACATG-3′), SM22α-Cre (5′-GCG GTC TGG CAG TAA AAA CTA TC-3′, 5′-GTG AAA CAG CAT TGC TGT CAC TT-3′, 5′-CTA GGC CAC AGA ATT GAA AGA TCT -3′, and 5′-GTA GGT GGA AAT TCT AGC ATC ATC C-3′), Apoe (5′-GCCTAGCCGAGGGAGAGCCG-3′, 5′-GTGACT TGGGAGCTCTGCAGC-3′, and 5′-GCCGCCCCGACTGCATCT-3′).

All animal procedures were performed under protocols approved by Yale University Institutional Animal Care and Use Committee.

## Vascular study

Animals were lightly anesthetized with inhaled isoflurane (0.2% in $O_2$). For aorta diameter measurement, B-mode ultrasound images of the ascending aorta in longitudinal plans were obtained using a Vevo 770 system (VisualSonics).

## Echocardiographic studies

Experiments were performed at the Yale Translational Research Imaging Center Core Facility. Cardiac function was analyzed by echocardiography using a Vevo 770 console (VisualSonics). Mice body temperature was maintained with a heading pad. Mice were anesthetized with 2% isoflurane, maintained under anesthesia with 1% isoflurane, and examined. The mouse was placed chest up on an examination board interfaced with the Vevo 770. Warmed Aquasonic gel was applied over the thorax and a 30-MHz probe was positioned over the chest in a parasternal position. Long- and short-axis B-mode and M-mode images were recorded. All measurements were obtained from three to six consecutive cardiac cycles, and the averaged values were used for analysis. Upon completion of the procedure, the gel was wiped off and the animal was returned to its cage housed in a warm chamber.

## Serum lipid analysis

Serum was obtained through centrifugation of the blood for 2 min at 8,944 *g* at 4°C and stored at −80°C until each assay was performed. Measurement of total cholesterol, triglycerides, and HDL cholesterol levels was performed in the Yale Mouse Metabolic Phenotyping Center.

## Histology and morphometric analysis

For human vessel studies, sections of left main coronary arteries were stained with elastic Van Gieson (EVG). Digital EVG-stained photographs of one section from each block were projected at final magnifications of 100×. ImageJ software (NIH) was used for morphometric analyses. As described in Fig 4B, measurements were made of the intima and media thickness. The ratio of intima (I) to media (M) thickness was used to grade the severity of atherosclerosis. The results for these parameters from each specimen were average of four different areas to obtain mean values. Left main coronary arteries of I/M ratio < 0.2 were considered as no disease or mild disease; those of I/M ratio between 0.2 and 1 were considered as moderate disease; those of I/M ratio greater than 1 or have calcification as severe disease (Chen *et al*, 2015).

## Histological analysis of atherosclerotic lesions

*Apoe*$^{-/-}$ and *Frs2*$^{SMCKO}$/*Apoe*$^{-/-}$ male mice were fed a Western diet (40% kcal% fat, 1.25% cholesterol, 0% cholic acid) for 8 or 16 weeks (Research Diets, product #D12108) starting at the age of 8 weeks. After 8 or 16 weeks of being fed a high-fat diet, mice were anesthetized and euthanized. Mouse heart was perfused with 10 ml of Dulbecco's phosphate-buffered saline (DPBS) (Life Technologies Cat #14190-144) and 10 ml of 4% paraformaldehyde (Polysciences, Inc. Cat #18814) via the left ventricle. To measure lesions in the

**The paper explained**

**Problem**

Atherosclerosis is a disease characterized by a slow build-up of plaque composed of living cell, cell debris, and extracellular matrix that leads to progressive narrowing and occlusion, thereby cutting off blood flow supply. It is the most common cause of heart attacks, stroke, and related complications. Anti-atherosclerosis therapy has been focused on reduction in cholesterol levels and various systemic anti-inflammatory treatments. A growth factor, transforming growth factor beta (TGFβ), is thought to play an important but ill-defined role in this process. In various studies, it has been reported as having pro- and anti-atherosclerotic activities and its effect in specific cellular context have not been carefully examined. Furthermore, little is known about what regulates this pathway.

**Results**

Here, we show that FGF regulates TGFβ signaling in smooth muscle cells. In cultured smooth cells *in vitro*, the loss of FGF signaling input leads to a dramatic reduction in *let-7* miRNA levels that, in turn, increase expression of TGFβ ligands and receptors and activation of TGFβ signaling. This, in turn, leads to growth arrest of proliferating smooth muscle cells and induction of their differentiation. A disruption of smooth muscle FGF signaling *in vivo* similarly leads to reduced proliferation of arterial smooth muscle cells. In atherosclerotic context, this results in a profound reduction in the size of atherosclerotic plaques. Finally, analysis of clinical specimens confirms the inverse relationship between the extent of medial FGF and TGFβ signaling and the severity of atherosclerosis.

**Impact**

These results demonstrate the critical role played by the FGF/TGFβ cross talk in regulation of smooth muscle phenotype switching and emphasize the importance of smooth muscle cell proliferation to growth and progression of atherosclerotic lesions. Taken together with the recent demonstration of deleterious effects of activation of endothelial TGFβ signaling, these results point to cell-type specific effects of TGFβ signaling and identify this pathway as a novel therapeutic intervention point.

aorta, the whole aorta, including the ascending arch, thoracic, and abdominal segments, was dissected, gently cleaned of adventitial tissue, and stained with Oil Red O (Sigma O0625) as previously described (Chen *et al*, 2015). The surface lesion area was quantified with ImageJ software (NIH).

## Immunohistochemical staining

Blocks were sectioned at 5-μm intervals using a Microm cryostat (for frozen blocks) or a Paraffin Microtome (for paraffin blocks). For frozen tissue sections, slides were fixed in acetone for 10 min at −20°C. For paraffin sections, slides were dewaxed in xylene, boiled for 20 min in citrate buffer (10 mM, pH 6.0) for antigen retrieval, and rehydrated. After washing three times with phosphate-buffered saline, tissue sections were incubated with primary antibodies diluted in blocking solution (10% BSA and horse serum in PBS) overnight at 4°C in a humidified chamber. For p-Smad2 and p-Smad3 staining, slides were denatured with 1.5 M HCl for 20 min prior to antibody labeling.

Sections were washed three times with Tris-buffered saline, incubated with appropriate Alexa Fluor 488-, Alexa Fluor 594-, or Alexa

Fluor 647-conjugated secondary antibodies diluted 1:1,000 in blocking solution for 1 h at room temperature, washed again three times, and mounted on slides with ProLong Gold mounting reagent with DAPI (Life Technologies P36931). All immunofluorescence micrographs were acquired using a Zeiss microscope. Images were captured using Velocity software and quantifications performed using ImageJ software (NIH).

## Statistical analysis

All graphs were created using GraphPad Prism software, and statistical analyses were calculated using GraphPad Prism. The significance of the differences between the controls and the experimental groups was determined using a two-tailed Student's *t*-test. For multiple comparisons, one-way ANOVA with Newman–Keuls test or chi-square test was used. A *P*-value < 0.05 was considered significant (*$P < 0.05$, **$P < 0.01$, ***$P < 0.001$). All results were confirmed by at least three independent experiments. A full table of *P*-values for figures is shown in Appendix Table S1.

## Study approval

All experiments involving animals were reviewed and approved by the Animal Welfare Committee of Yale University. The ethics committee of Yale University approved the procedures related to human subjects. All patients who participated in the study provided written informed consent.

Expanded View for this article is available online.

## Acknowledgements

We are grateful to Rita Webber and Nicole Copeland for maintaining mice colonies used in this study. This work was supported by NIH Grant R01 HL053793 (M.S.). Blood lipid profile was done by the Yale Mouse Metabolic Phenotyping Center (MMPC) core facility (NIH Grant U24 DK059635).

## Author contributions

P-YC, LQ, GL, GT, and MS designed research; P-YC, LQ, and GL performed research; P-YC, LQ, GL, GT, and MS analyzed data; and P-YC and MS wrote the manuscript. All authors read and approved the manuscript.

## Conflict of interest

The authors declare that they have no conflict of interest.

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
