## [Review Process File · EMBO Molecular Medicine]

Smooth muscle FGF/TGF cross-talk regulates atherosclerosis progression

Pei-Yu Chen, Lingfeng Qin, Guangxin Li, George Tellides and Michael Simons

Corresponding author: Michael Simons and Pei-Yu Chen, Yale University School of Medicine

Review timeline:

Submission date:	31 December 2015
Editorial Decision:	04 February 2016
Revision received:	15 March 2016
Editorial Decision:	05 April 2016
Revision received:	07 April 2016
Accepted:	11 April 2016

Transaction Report:

Editor: Roberto Buccione

1st Editorial Decision

04 February 2016

Thank you for the submission of your Report manuscript to EMBO Molecular Medicine. We have now received comments from the two Reviewers whom we asked to evaluate your manuscript

You will see that the Reviewers are quite supportive of your work, although they do raise a few issues that prevent us from considering publication at this time. I will not dwell into much detail, as the evaluations are self-explanatory.

Reviewer 2 would like you to convincingly show that TGF β is directly involved in the decreased proliferation of the FRS2a KO SMCs and also requires better images, explanation of discrepancies in protein size in the blots, provision of quantification for the western blots and better description of the genetic make-up of the animals (and clarification on controls). Finally s/he would like you to discuss the therapeutic implications of your work.

In conclusion, while publication of the paper cannot be considered at this stage, we would be pleased to consider a suitably revised submission, provided that the above concerns are addressed.

Please note that it is EMBO Molecular Medicine policy to allow a single round of revision only and that, therefore, acceptance or rejection of the manuscript will depend on the completeness of your responses, as outlined above, included in the next, final version of the manuscript.

As you know, EMBO Molecular Medicine has a "scooping protection" policy, whereby similar findings that are published by others during review or revision are not a criterion for rejection. Although I clearly do not foresee such an instance in this case, I do ask you to get in touch with us after three months if you have not completed your revision, to update us on the status. Please also

contact us as soon as possible if similar work is published elsewhere.

Please note that EMBO Molecular Medicine now requires a complete author checklist (<http://embomolmed.embopress.org/authorguide#editorial3>) to be submitted with all revised manuscripts. Provision of the author checklist is mandatory at revision stage; The checklist is designed to enhance and standardize reporting of key information in research papers and to support reanalysis and repetition of experiments by the community. The list covers key information for figure panels and captions and focuses on statistics, the reporting of reagents, animal models and human subject-derived data, as well as guidance to optimise data accessibility. The Author checklist will be published alongside the paper, in case of acceptance, within the transparent review process file.

I also suggest that you carefully adhere to our guidelines for publication in your next version and especially our new requirements for supplemental data (see also below) to speed up the pre-acceptance process.

I look forward to seeing a revised form of your manuscript as soon as possible.

***** Reviewer's comments *****

Referee #1 (Remarks):

The manuscript by Pei-Yu Chen and colleagues provides three lines of evidence supporting a relationship between FGF signaling (through FRS2a) and TGF β signaling during the development of atherosclerotic plaques and the contractile/proliferative switch in smooth muscle cells. It is a complete work.

Briefly, in cultured cells they manipulated expression of FRS2a and measured the effect on TGF β signaling (a few transcripts and a key protein modifications) and differentiation status (a few transcripts and a couple of phenotypic assay/responses). Importantly, the notion of the effect of TGF β signaling changes as a driver of these effects were followed up by manipulating TGF β signaling, both pharmacologically and by using two different shRNA manipulations at different levels of the TGF β pathway.

Complementing this work they also showed that knocking down FGFR1 would give a similar effect as that of knocking down FRS2a (albeit I would have liked to see the phenotypic studies as well-and why is the y-axis is Supplementary figure 2A different than any other 'relative mRNA expression'). Continuing this, they demonstrated that enforced expression of let7b, a downstream target repressed by FGF signaling loss, prevented the activation of TGF β (some immunoblot assays). A look at the timing of expression and phenotypic changes provides evidence consistent with this notion as well.

This work is followed by an analysis of marker expression in human tissue and causality is established in a mouse model.

Referee #2 (Remarks):

The authors investigated the effect of SMC-specific suppression of FGF signaling on TGF β -induced SMC differentiation/proliferation and atherosclerotic progression. This study is an extension of the previous studies by the same group, in which they showed EC-specific deletion of Frs2 α caused activation of TGF β signaling, resulting in an increase in Endo-MT and enhancement of atherosclerosis in Apoe $^{-/-}$ mice. In the current study, the authors employed a similar strategy in vitro and in vivo and showed that deletion of Frs2 in SMCs exhibited the opposite effect on atherosclerotic progression compared to Frs2ECKO; Apoe $^{-/-}$ mice. This is an interesting paper; however, it lacks mechanistic novelty. The link between suppression of FGF and increased TGF β signaling (mediated by let-7 miRNA) has already been established. Therapeutically, is suppression

of FGF signaling in the aortic wall effective for atherosclerosis?

Specific comments:

1. The authors convincingly showed that knockdown (KD) of FRS2alpha led to the increased differentiation of SMCs in a TGFbeta-dependent manner. However, the evidence that TGFbeta is indeed involved in the decreased proliferation of FRS2alpha-KD SMCs is not presented. The authors should provide the evidence by using SB, shTGFbR2 or shSmad2.
2. Discussion on overall therapeutic strategy against atherosclerosis involving FGF/TGFbeta signaling pathways should be provided.
3. In Figures 6, 4-month HFD sections seem morphologically different between 6C-D and 6E-F; therefore, it is difficult to compare the localization and intensity of p-FGFR1, p-Smad2 and p-Smad3.
3. It seems that the band size fluctuates in FRS2alpha blots in Figures 2D-F and Figure 3E. Any explanations?
4. Only representative image are provided for all Western analyses without quantification. Some blots are difficult to appreciate the differences without quantification (Figure S2E). Please provide.
5. What is the genetic background of SM22alpha-Cre; Frs2flox/flox; Apoe-/- and Apoe-/-? Did the authors use littermates for control?

1st Revision - authors' response

15 March 2016

We appreciate the reviewers' positive comments on our manuscript EMM-2015-06181. The specific points raised by the reviewers are addressed as follows (The original reviewer comments are in italics and our replies are in the regular font):

Referee #1 (Remarks):

The manuscript by Pei-Yu Chen and colleagues provides three lines of evidence supporting a relationship between FGF signaling (through FRS2a) and TGFb signaling during the development of atherosclerotic plaques and the contractile/proliferative switch in smooth muscle cells. It is a complete work.

Briefly, in cultured cells they manipulated expression of FRS2a and measured the effect on TGFb signaling (a few transcripts and a key protein modifications) and differentiation status (a few transcripts and a couple of phenotypic assay/responses). Importantly, the notion of the effect of TGFb signaling changes as a driver of these effects were followed up by manipulating TGFb signaling, both pharmacologically and by using two different shRNA manipulations at different levels of the TGFb pathway.

Complementing this work they also showed that knocking down FGFR1 would give a similar effect as that of knocking down FRS2a (albeit I would have liked to see the phenotypic studies as well-and why is the y-axis in Supplementary figure 2A different than any other 'relative mRNA expression'). Continuing this, they demonstrated that enforced expression of let7b, a downstream target repressed by FGF signaling loss, prevented the activation of TGFb (some immunoblot assays). A look at the timing of expression and phenotypic changes provides evidence consistent with this notion as well.

This work is followed by an analysis of marker expression in human tissue and causality is established in a mouse model.

Reply: We appreciate the positive assessment of the study.

General Comments:

1. *Complementing this work they also showed that knocking down FGFR1 would give a similar effect as that of knocking down FRS2 α (albeit I would have liked to see the phenotypic studies as well).*

Reply: We provide new results (Figure EV2) showing that FGFR1 knockdown under growth condition induces SMC differentiation similar to FRS2 α knockdown and that inhibition of TGF β signaling using a variety of means (SB431542, TGF β R2 shRNA, and Smad2 shRNA) reverses this effect (Figure EV2 A-C; compare to Figure 2 D-F).

We further show the effects of FGFR1 knockdown and the *let-7b* miRNA rescue under differentiation condition (Figure EV2 D-E): these are similar to FRS2 α results shown in Figure 3E.

2. *why is the y-axis is Supplementary figure 2A different than any other 'relative mRNA expression'.*

Reply: Supplementary Figure 2A is now Appendix Figure S2A.

In these experiments we were comparing FGFR1, FGFR2, FGFR3, and FGFR4 expression in HUVEC. We are using mRNA copy number to show different FGFR expression abundance.

Referee #2 (Remarks):

The authors investigated the effect of SMC-specific suppression of FGF signaling on TGFbeta-induced SMC differentiation/proliferation and atherosclerotic progression. This study is an extension of the previous studies by the same group, in which they showed EC-specific deletion of Frs2alpha caused activation of TGFbeta signaling, resulting in an increase in Endo-MT and enhancement of atherosclerosis in Apoe-/- mice. In the current study, the authors employed a similar strategy in vitro and in vivo and showed that deletion of Frs2 in SMCs exhibited the opposite effect on atherosclerotic progression compared to Frs2ECKO; Apoe-/- mice. This is an interesting paper; however, it lacks mechanistic novelty. The link between suppression of FGF and increased TGFbeta signaling (mediated by let-7 miRNA) has already been established. Therapeutically, is suppression of FGF signaling in the aortic wall effective for atherosclerosis?

Specific comments:

1. *The authors convincingly showed that knockdown (KD) of FRS2alpha led to the increased differentiation of SMCs in a TGFbeta-dependent manner. However, the evidence that TGFbeta is indeed involved in the decreased proliferation of FRS2alpha-KD SMCs is not presented. The authors should provide the evidence by using SB, shTGFbR2 or shSmad2.*

Reply: We provided new evidence (Figure EV1) that inhibition of TGF β signaling (SB431542, shTGF β R2, shSmad2) reversed FRS2 α knockdown-induced cell cycle arrest using a number of different assays including WST-1 (water-soluble tetrazolium salt) Cell Proliferation Assay, BrdU immunofluorescence assay, and BrdU colorimetric cell proliferation ELISA.

We also added the WST-1 Cell Proliferation and BrdU assays to the Materials and Methods section.

2. *Discussion on overall therapeutic strategy against atherosclerosis involving FGF/TGFbeta signaling pathways should be provided.*

Reply: This has been added to the Discussion section (page 15).

3. In Figures 6, 4-month HFD sections seem morphologically different between 6C-D and 6E-F; therefore, it is difficult to compare the localization and intensity of p-FGFR1, p-Smad2 and p-Smad3.

Reply: We stained normal diet and 4M HFD brachiocephalic artery serial sections with p-FGFR1, FGFR1, p-Smad2, and p-Smad3 antibodies and compared the positive cell numbers in the medial smooth muscle cell layers. We replaced Figure 6 C-F with new images.

4. *It seems that the band size fluctuates in FRS2alpha blots in Figures 2D-F and Figure 3E. Any explanations?*

Reply: FRS2 α protein has 6 tyrosine residues and more than 30 serine/threonine residues. It often shows different electrophoretic mobility shifts caused by phosphorylation on serine/threonine residues.

For example, see

Lax et al., Mol. Cell 10: 709-719, 2002.

Wu et al., Biol Chem. 384: 1215-1226, 2003.

5. *Only representative image are provided for all Western analyses without quantification. Some blots are difficult to appreciate the differences without quantification (Figure S2E). Please provide.*

Reply: We have performed Western blot quantification in all of our figures (except Figure 3E, Figure EV2E, and Figure EV3B) including Figure 1C, Figure 2A, Figure 2D-F, Figure 3B, Figure EV2A-C, Appendix Figure S1B, and Appendix Figure S2E.

We also added the *Quantification of Western blots* in the Materials and Methods.

6. *What is the genetic background of SM22alpha-Cre; Frs2flox/flox; Apoe^{-/-} and Apoe^{-/-}? Did the authors use littermates for control?*

Reply: All mice are on the C57BL/6 background. This has been added to the Materials and Methods- Generation of mice section.

Frs2^{flox/flox}, Apoe^{-/-} littermates were used as controls.

Thank you for the submission of your revised manuscript to EMBO Molecular Medicine. We have now received the enclosed reports from the referees that were asked to re-assess it. As you will see the reviewers are now globally supportive and I am pleased to inform you that we will be able to

accept your manuscript pending the following final minor amendments:

1) Could you please collect the P values now featured in the source data files into a single appendix table? This would also imply inserting the appropriate callouts in the manuscript in the figure legends where applicable. Sorry for this added hassle but the goal is to make the information more readily accessible. This will also imply removing altogether the source data files for figures 4, 5, 6, 7 and the p values table from source data file 3, as they would be no longer needed.

2) Every published paper now includes a 'Synopsis' to further enhance discoverability. Synopses are displayed on the journal webpage and are freely accessible to all readers. They include a short standfirst as well as 2-5 one sentence bullet points that summarise the paper. Please provide the synopsis including the short list of bullet points that summarise the key NEW findings. The bullet points should be designed to be complementary to the abstract - i.e. not repeat the same text. We encourage inclusion of key acronyms and quantitative information. Please use the passive voice. Please attach this information in a separate file or send them by email, we will incorporate it accordingly. You are also welcome to suggest a striking image or visual abstract to illustrate your article. If you do please provide a jpeg file 550 px-wide x 400-px high.

Please submit your revised manuscript within two weeks. I look forward to seeing a revised form of your manuscript as soon as possible.

***** Reviewer's comments *****

Referee #1 (Comments on Novelty/Model System):

This is a well done study and has only been improved by the authors changes.

Referee #1 (Remarks):

This was a nice comprehensive manuscript before review and the additional data added improved it further.

Referee #2 (Remarks):

Previous concerns were addressed adequately.

2nd Revision - authors' response

07 April 2016

We are grateful to the reviewers and the Editor for a thorough review on our manuscript EMM-2015-06181. The comments and our replies follow below.

Comments from the Editor:

1) Could you please collect the P values now featured in the source data files into a single appendix table? This would also imply inserting the appropriate callouts in the manuscript in the figure legends where applicable. Sorry for this added hassle but the goal is to make the information more readily accessible. This will also imply removing altogether the source data files for figures 4, 5, 6, 7 and the p values table from source data file 3, as they would be no longer needed.

Reply: We collected all the p-values into a single Appendix Table S1. We also reference these items in text, figure legends, and materials and methods-statistical analysis section.

Source data files for Figures 4, 5, 6, 7 and EV Figures 1, 4, 5 are removed. p-value tables from source data Figures 1, 2, 3, EV Figure 3, Appendix Figure S1, and Appendix Figure S2 are deleted.

2) Every published paper now includes a 'Synopsis' to further enhance discoverability. Synopses are displayed on the journal webpage and are freely accessible to all readers. They include a short standfirst as well as 2-5 one sentence bullet points that summarise the paper. Please provide the synopsis including the short list of bullet points that summarise the key NEW findings. The bullet points should be designed to be complementary to the abstract - i.e. not repeat the same text. We encourage inclusion of key acronyms and quantitative information. Please use the passive voice. Please attach this information in a separate file or send them by email, we will incorporate it accordingly. You are also welcome to suggest a striking image or visual abstract to illustrate your article. If you do please provide a jpeg file 550 px-wide x 400-px high.

Reply: We have the Synopses summary in a separate word document; we also provide four options of Synopses jpeg images.

Comments from the Reviewers:

Referee #1 (Comments on Novelty/Model System):

This is a well done study and has only been improved by the authors changes.

Reply: Ok.

Referee #1 (Remarks):

This was a nice comprehensive manuscript before review and the additional data added improved it further.

Reply: Thank you.

Referee #2 (Remarks):

Previous concerns were addressed adequately.

Reply: Thank you.

Corresponding Author Name: Prof. Michael Simons

Manuscript Number: EMM-2015-06181